# Overemphasized role of preceding strong El Niño in generating multi-year La Niña events

Ji-Won Kim [1] ✉, Jin-Yi Yu [2] ✉ & Baijun Tian [1]

Previous studies have emphasized the significance of a strong El Niño preceding La Niña (LN) in the formation of multi-year LN events due to the slow recharge-discharge ocean heat content process. However, observational analyses from 1900 to 2022 reveal that the majority (64%) of multi-year LN events did not necessitate a preceding strong El Niño to generate their second LN, suggesting an overemphasis on traditional views. Instead, here we show that a negative phase of the North Pacific Meridional Mode (PMM) during spring, when the first LN begins to decay, activates the mechanism responsible for triggering another LN and producing a multi-year event. The westward extension of the first LN's cold anomalies, which interact directly with the eastern edge of the western Pacific warm pool, is highlighted as a crucial factor in the occurrence of a negative PMM. Additionally, the PMM mechanism can create a third LN, leading to triple-dip events.

El Niño and La Niña (LN) events, the warm and cold phases of the El Niño/Southern Oscillation (ENSO), respectively, exhibit considerable inter-event differences with respect to their amplitudes, spatial structures, and temporal evolutions (known as complex ENSO behaviors)[1-8]. One of the complex ENSO behaviors arises from the diverse temporal evolutions, which can be characterized as transitional single-year or successive multi-year ENSO events[9-18]. For example, some ENSO events rapidly decay after their peak and transition to the opposite phase in the following year, resulting in a transitional single-year event. Others, however, do not decay after their peak but linger or re-intensify in the following year, producing a successive multi-year event. The multi-year event can be further divided into a lingering or re-intensified type according to its decaying evolution pattern[19]. Understanding the diverse ENSO evolution patterns and their underlying mechanisms has been a central topic for ENSO research in the past decade, as the ability to precisely predict whether an ENSO event will terminate shortly after 1 year or last 2 years or longer is of global importance.

Unlike most El Niño events, which typically have a short lifetime within a year, roughly more than half of LN events (43–70%)[16-18] persist for 2 years or longer, becoming a multi-year LN[20-25]. There have also been increased occurrences of LN events that persist for multiple years since the 1990s, including not only double-dip LN (events that last 2 years) but also triple-dip LN (events that last 3 years), like the latest

event that began in September 2020[17,26] (Supplementary Table 1). Studies have shown that these multi-year LN events can be particularly impactful, as they exert more severe climate impacts or shifts in anomaly patterns than single-year LN events. For instance, they cause consecutive droughts in the southern United States during boreal winter[27], extreme warm temperatures over East Asian countries during boreal summer[28,29], torrential rain and floods in eastern Australia during austral winter and spring[30], and pronounced zonal shifts in Antarctic sea ice concentration anomalies during austral winter[31]. Therefore, researchers have focused on gaining a better understanding of the dynamics that govern the temporal evolution of LN, with a greater emphasis on studying the mechanisms behind the formation of multi-year LN events.

One of the primary mechanisms is based on the slow recharge-discharge process of ocean heat content (OHC), which is reflected by sea surface height or thermocline depth anomalies in the equatorial Pacific[32-34]. This OHC mechanism creates long-term oceanic memory across ENSO phases, carrying the effect of a strong El Niño from the preceding year into subsequent LN events. Specifically, a strong El Niño amplitude in the year prior to an LN can greatly deplete the equatorial band of upper-ocean heat content, leading to a significant heat deficit (or discharge) in the equatorial Pacific. The depletion is induced by large-scale off-equatorial easterly anomalies during the

[1]Jet Propulsion Laboratory, California Institute of Technology, Pasadena, CA, USA. [2]Department of Earth System Science, University of California, Irvine, CA, USA. ✉e-mail: jiwon.kim@jpl.nasa.gov; jyyu@uci.edu

decay stage of the preceding strong El Niño, which efficiently drives meridional Ekman heat transport away from the equator[18]. The large heat discharge resulting from the strong El Niño takes years to be restored owing to the slower heat build-up/recharge process by the subsequent LN[22,24], providing conditions for the LN to persist after 1 year, giving rise to the development of a multi-year event. Motivated by this, previous studies have highlighted that a preceding strong El Niño is a critical factor in the formation of multi-year LN events[16–18,22,35–37].

Meanwhile, some recent studies have increasingly recognized that multi-year LN events also exhibit a robust connection to mid-latitudes through the North Pacific Meridional Mode (PMM, simply)[38]. During the developing phase of a first LN in boreal spring, a negative phase of the PMM often emerges as a result of the Gill-type atmospheric response to strong El Niño sea surface temperature (SST) anomalies over the equatorial eastern Pacific[10,39]. The negative PMM endures in subsequent seasons due to thermodynamic air-sea couplings over the subtropical North Pacific, favoring the development of an LN event in winter, characterized by a broader meridional pattern of cold SST and easterly anomalies[13,18]. This meridionally broad LN is accompanied by a weaker negative wind stress curl at more extratropical latitudes, leading to a slower heat recharge of the equatorial Pacific, which enables the cold anomalies to persist, ultimately resulting in a multi-year event[10,13,17,18,40,41]. While this mid-latitude connection offers an alternative pathway for the formation of multi-year LN events, it still necessitates a preceding strong El Niño as a key component.

In this study, we analyze various observational data covering the 20th century (1900–2022) and demonstrate that the role of preceding strong El Niño has been overemphasized, as the majority of multi-year LN events did not actually require a strong El Niño in the preceding year. Instead, we identify that the mechanism associated with a negative PMM following the first LN, which involves two-way interactions between tropical ENSO and subtropical PMM, plays a crucial role in these events. Our proposed mechanism, termed the PMM mechanism, can operate independently without the need for the existence of a preceding strong El Niño and is applicable not only to double-dip but also to triple-dip LN events. Consequently, it provides a comprehensive explanation for the generation of multi-year LN events, irrespective of the El Niño conditions in their preceding years.

## Results
### Role of preceding strong El Niño: Overemphasized
We identified 22 LN events that occurred between 1900 and 2022 and classified them into three categories: multi-year, single-year, or neither (Table 1; see 'Identifying La Niña events and their classification' in Methods). As a result, 11 of the LN events were multi-year, 6 were single-year, and 5 were neither. The five LN events that were neither multi-year nor single-year events, due to their neutral ENSO condition in the decaying year, were excluded from the analysis. This exclusion allows for a clearer contrast in evolution characteristics between multi-year and single-year LN events (Supplementary Fig. 1).

The Niño3.4 index evolutions of individual multi-year LN events are presented in Fig. 1a, and they exhibit consistent characteristics from the boreal summer of the first year (Year 0) onwards (see 'Definition of climate indices' in Methods; hereafter, seasons will follow those in the Northern Hemisphere). There is no phase transition after the peak of the first LN, and the same phase is maintained during subsequent seasons of the second year (Year 1), re-intensifying to create a second LN. Contrastingly, the winters preceding the onset of the first LN show varied ENSO conditions, as seen in the large spread of the Niño3.4 index, ranging from −0.17 °C to 2.37 °C (Fig. 1a and Table 1). In Fig. 1b, for multi-year LN events (black dots), the spread of the Niño3.4 index during the preceding winter (November$^{-1}$–January$^{0}$) is ±0.95 °C (horizontal error bar across the black square), which is roughly three times larger than the spread of ±0.31 °C during their second

winter (November$^{1}$–January$^{2}$) (vertical error bar). Similar, but weaker, results are found for single-year LN events (gray dots), with the Niño3.4 index spread during the preceding winter being ±0.64 °C, which is about 1.8 times larger than that during the second winter (±0.36 °C) (error bars across the gray square). Furthermore, it is seen that only 4 out of 11 (36%) multi-year LN events (black dots with red outline) are preceded by a strong El Niño, defined as having Niño3.4 indices >1.5 °C. The remaining 7 events (64%) (black dots with blue outline) have Niño3.4 indices <1.5 °C (and even less than 0.9 °C), indicating that a majority of the multi-year LN events did not necessitate a preceding strong El Niño. These findings challenge the current understanding and raise the question: Is a preceding strong El Niño required to generate multi-year LN events?

To investigate the importance of preceding ENSO conditions in the formation of multi-year LN events, we compare the Niño3.4 index evolutions among three groups of LN events in Fig. 1c: multi-year LN with a preceding strong El Niño (referred to as myLN_wPrSEN) and those without (referred to as myLN_w/oPrSEN), and single-year LN (referred to as syLN). As expected, the Niño3.4 index evolutions of myLN_wPrSEN and myLN_w/oPrSEN exhibit contrasting ENSO conditions only in the seasons preceding the onset of the first LN (red and blue curves). It is interesting that the preceding weak El Niño condition of myLN_w/oPrSEN is comparable to that of syLN (blue and gray curves), suggesting that something other than preceding ENSO conditions is responsible for these two LN groups developing into such dramatically different evolution patterns.

We examine the recharge-discharge process associated with the three LN groups by analyzing their OHC index evolutions (Fig. 1d). For myLN_wPrSEN (red curve), the preceding strong El Niño leaves a significant, large discharge of OHC (OHC index <−2 s.d.) along the equatorial Pacific, triggering the onset of the first LN. The negative OHC index rapidly increases through the summer and fall of the first year (June$^{0}$–November$^{0}$), coinciding with the development of the first LN. The subsequent recharge process by the first LN is insufficient to restore the OHC index to normal (OHC index = 0), thus maintaining it throughout the whole second year until the spring of the third year when the OHC index finally returns to zero. Therefore, the OHC index evolution of myLN_wPrSEN is consistent with the OHC mechanism mentioned earlier, which links a preceding strong El Niño to multi-year LN via a slow recharge-discharge process. However, this process cannot explain the formation of the other multi-year LN group, myLN_w/oPrSEN (blue curve). The preceding weak El Niño condition results in only a small discharge, which is quickly restored to normal through the recharge process during the second year. The OHC index evolution of myLN_w/oPrSEN closely resembles that of syLN from the preceding year onwards until the first LN decays in the spring of the second year (blue and gray curves). The difference between the OHC index evolution of myLN_w/oPrSEN and myLN_wPrSEN and its similarity to syLN can also be confirmed in Fig. 1e, which displays the seasonally averaged OHC intensity during June$^{0}$–November$^{0}$. On average, the OHC intensity of myLN_wPrSEN (−2.45 ± 1.11 s.d.) is ~5 times greater than the intensities of myLN_w/oPrSEN and syLN (−0.45 ± 0.51 s.d. and −0.48 ± 0.58 s.d., respectively).

In Fig. 1f, we further create a phase space diagram based on the Niño3.4 and OHC indices for the three LN groups to directly compare their time-varying phase transitions. The red curve, representing myLN_wPrSEN, shows that the preceding strong El Niño causes a large discharge, producing a first LN (from circle to square) that slowly recovers but remains in the negative phase until the peak of the second LN (from square to triangle). This supports the OHC mechanism. However, the weak OHC phase transitions from the preceding weak El Niño until the peak of the first LN are evident in the blue and gray curves representing myLN_w/oPrSEN and syLN, respectively. During the subsequent recharge process by the first LN, the discharged OHC easily returns to a neutral phase. However, for myLN_w/oPrSEN, a

**Table 1 | Classification of La Niña (LN) events**

| La Niña classification | | Event year | Preceding winter Niño3.4 index | First winter Niño3.4 index | Second winter Niño3.4 index | Third winter Niño3.4 index | Fourth winter Niño3.4 index |
|---|---|---|---|---|---|---|---|
| Multi-year La Niña | w/ a preceding strong El Niño | 1973–1975(T) | 1.87 | **−2.00** | **−0.69** | **−1.58** | 0.80 |
| | | 1983–1985(D) | 2.26 | **−0.85** | **−1.13** | −0.49 | - |
| | | 1998–2000(T) | 2.37 | **−1.31** | **−1.51** | **−0.78** | −0.27 |
| | | 2010–2012(D) | 1.56 | **−1.57** | **−0.97** | −0.03 | - |
| | w/o a preceding strong El Niño | 1908–1910(T) | −0.17 | **−0.73** | **−1.13** | **−0.54** | 1.13 |
| | | 1916–1918(D) | −0.14 | **−1.59** | **−0.63** | 1.38 | - |
| | | 1949–1951(D) | −0.10 | **−1.13** | **−1.13** | 0.64 | - |
| | | 1954–1956(T) | 0.30 | **−0.84** | **−1.50** | **−0.68** | 1.26 |
| | | 1970–1972(D) | 0.68 | **−1.10** | **−0.75** | 1.87 | - |
| | | 2007–2009(D) | 0.90 | **−1.58** | **−0.69** | 1.56 | - |
| | | 2020–2022(T) | 0.65 | **−0.95** | **−0.85** | **−0.74** | N/A |
| Single-year La Niña | | 1903/1904 | 1.43 | **−0.79** | 0.60 | - | - |
| | | 1924/1925 | 0.78 | **−0.88** | 1.37 | - | - |
| | | 1938/1939 | 0.11 | **−0.69** | 0.55 | - | - |
| | | 1964/1965 | 0.86 | **−0.87** | 1.38 | - | - |
| | | 2005/2006 | 0.65 | **−0.67** | 0.90 | - | - |
| | | 2017/2018 | −0.42 | **−0.78** | 0.84 | - | - |
| Neither La Niña | | 1921/1922 | 0.14 | **−0.65** | −0.38 | - | - |
| | | 1933/1934 | −0.18 | **−1.01** | −0.18 | - | - |
| | | 1942/1943 | 1.04 | **−1.26** | −0.40 | - | - |
| | | 1988/1989 | 1.04 | **−1.93** | −0.10 | - | - |
| | | 1995/1996 | 1.14 | **−0.65** | −0.34 | - | - |

Specific years of the observed 22 LN events from 1900 to 2022 and their corresponding Niño3.4 index values (in °C) during preceding, first, second, third, and fourth winters. LN events are classified as either multi-year (occurring with or without a preceding strong El Niño), single-year, or neither. The years with (T) denote a triple-dip event, while those with (D) denote a double-dip event. Values in bold (italics) indicate LN (El Niño) conditions with Niño3.4 index <−0.5 °C (>0.5 °C). Note that the data regarding potential El Niño conditions in the winter of 2023, following the 2020–2022 triple-dip LN, are not yet available.

second LN develops and moves the OHC index back to the discharge phase, forming a circular trajectory in the diagram. In contrast, for syLN, the discharged OHC transitions to the recharge phase and produces an El Niño in the second year. Collectively, these results suggest that the emphasis placed on the role of preceding strong El Niño has been excessive since the OHC mechanism only explains the behavior of the minor multi-year LN group, myLN_wPrSEN, while it cannot account for the major multi-year LN group, myLN_w/oPrSEN.

**Role of negative PMM: crucial**
Several studies have proposed that the subtropical North Pacific coupling processes associated with the PMM provide distinct opportunities to influence tropical Pacific variability[3–5,42,43]. Here we identify that the PMM mechanism (emphasizing its crucial role) is more important than the OHC mechanism in generating multi-year LN events. The PMM mechanism not only influences tropical Pacific variability and triggers ENSO events but can also be triggered by ENSO events[9,10,44,45], leading to another ENSO event with the same phase and resulting in a multi-year event[9–12,17,46].

In both multi-year LN groups, myLN_wPrSEN and myLN_w/oPrSEN, we observe that in the second spring after their first LN peaks (March[1]–May[1]), a band of cold SST anomalies accompanied by northeasterly anomalies extends southwestward from the subtropics to the equator (Fig. 2a, b; see parallelograms). This feature contains a defining characteristic of a negative PMM, as reflected in the leading ocean–atmosphere coupled mode between SST and wind anomalies over the subtropical North Pacific[38]. The PMM index evolutions (Fig. 2d) and the seasonally averaged PMM intensity during March[1]–May[1] (Fig. 2e) offer evidence of the occurrence of a negative PMM, as both multi-year LN groups exhibit a strong negative PMM index (PMM index <−1 s.d.). By contrast, the single-year LN group, syLN, does not show the features

associated with a negative PMM. This is evident from the absence of cold SST and northeasterly anomalies over the subtropical North Pacific (Fig. 2c; see parallelogram), as well as the near-neutral PMM phase and intensity from the second spring onwards (Fig. 2d, e). These results imply that the presence of a negative PMM during the second spring is critical in differentiating between multi-year and single-year LN events. The scatter plot in Fig. 2f corroborates this, as it shows that nearly all multi-year LN events (82%, 9 out of 11 events) in both myLN_wPrSEN and myLN_w/oPrSEN groups have PMM indices close to or less than −1 s.d. (black square and vertical error bar). Conversely, all single-year LN events have PMM indices >−1 s.d. (gray square and vertical error bar). This contrast in the PMM index, however, is not as noticeable in the OHC index because nearly all LN events, except for the myLN_wPrSEN events, have OHC indices within ±1 s.d. (black and gray squares and their horizontal error bars). It is worth mentioning that only two myLN_wPrSEN events occurred in 1983–1985 and 1998–2000, both of which were preceded by extreme El Niño events (with Niño3.4 index being >2 °C; Table 1 and Fig. 1a, b), had considerably lower OHC index values than other events. This may suggest that the OHC mechanism is only effective in these two specific events.

To understand why a negative PMM is more likely to occur during multi-year LN events than during single-year LN events, we need to identify the key difference between these two types. Our analysis finds that the longitudinal location of the first LN's cold anomalies is a crucial factor controlling the occurrence of a negative PMM. Evidence supporting this is presented in Fig. 3, which illustrates the longitude-time plots of SST and wind anomalies for the three LN groups, covering the tropical (5°S–5°N) and subtropical (15°N–25°N) Pacific basins. Despite the contrasting preceding El Niño conditions and different development structures, myLN_wPrSEN and myLN_w/oPrSEN (Fig. 3a, b) exhibit a shared characteristic during the mature winter of their first LN:

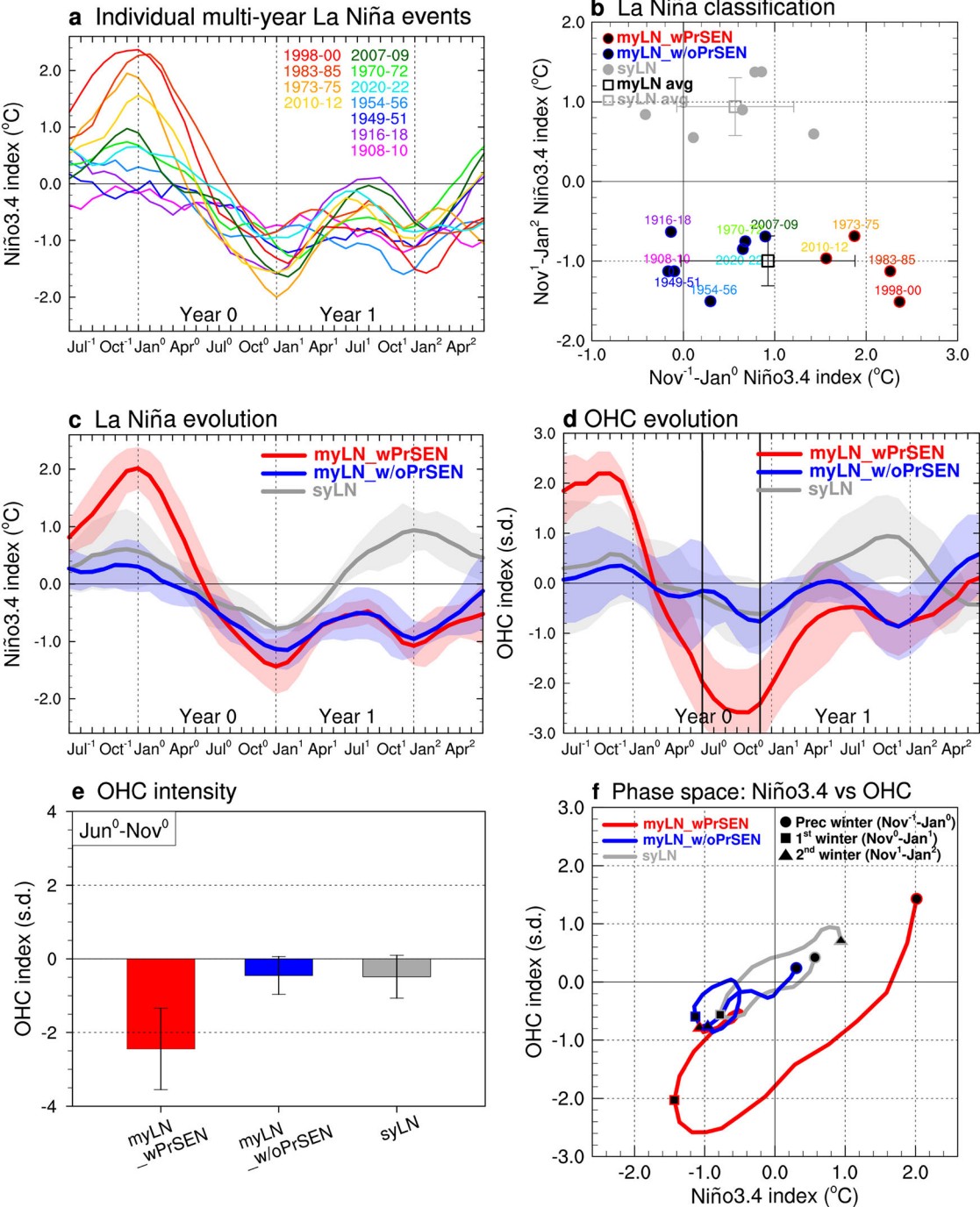

**Fig. 1 | Role of preceding strong El Niño in generating multi-year La Niña (LN). a** Evolution of Niño3.4 index for individual multi-year LN events during 1900–2022. Each event year is depicted with a different color, corresponding to its specific evolution. **b** Scatter plot of Niño3.4 index values during the preceding winter against those during the second-winter for multi-year LN events with a preceding strong El Niño (myLN_wPrSEN, black dots with red outline) or without (myLN_w/oPrSEN, black dots with blue outline) and single-year LN events (syLN, gray dots). The black (gray) square represents the average for multi (single)-year LN events with error bars indicating ±1 s.d. **c** Composite evolution of Niño3.4 index for the three LN groups: myLN_wPrSEN (red curve), myLN_w/oPrSEN (blue curve), and syLN (gray curve). Shaded areas represent their ±1 s.d. **d** Same as (**c**) except for the ocean heat content (OHC) index. **e** Bar-charts displaying the OHC intensity during June0–November0 for the three LN groups, with error bars indicating ±1 s.d. **f** Phase space diagram showing the Niño3.4 and OHC indices for myLN_wPrSEN (red curve), myLN_w/oPrSEN (blue curve), and syLN (gray curve) during the preceding (circle), first (square), and second (triangle) winters. The indices are smoothed with a 3-month running-mean filter.

westward-extended cold anomalies across the eastern edge of the western Pacific warm pool, where atmospheric deep convection is vigorously active (marked by the 28°C isotherm line; see green contour). These westward-extended cold anomalies can promptly induce strong negative diabatic heating anomalies in the tropical central Pacific (5°S–5°N, 165°E–165°W) by effectively reducing SSTs below the

convective threshold temperature[47]. Consequently, pronounced atmospheric suppression (or negative precipitation anomalies) occurs in that region (Supplementary Figs. 2a, b and 3a). This suppression readily excites stationary atmospheric Rossby waves into the North Pacific[10,44,45], inducing a subtropical anticyclone anomaly during the transition season between the first winter and the second spring, from

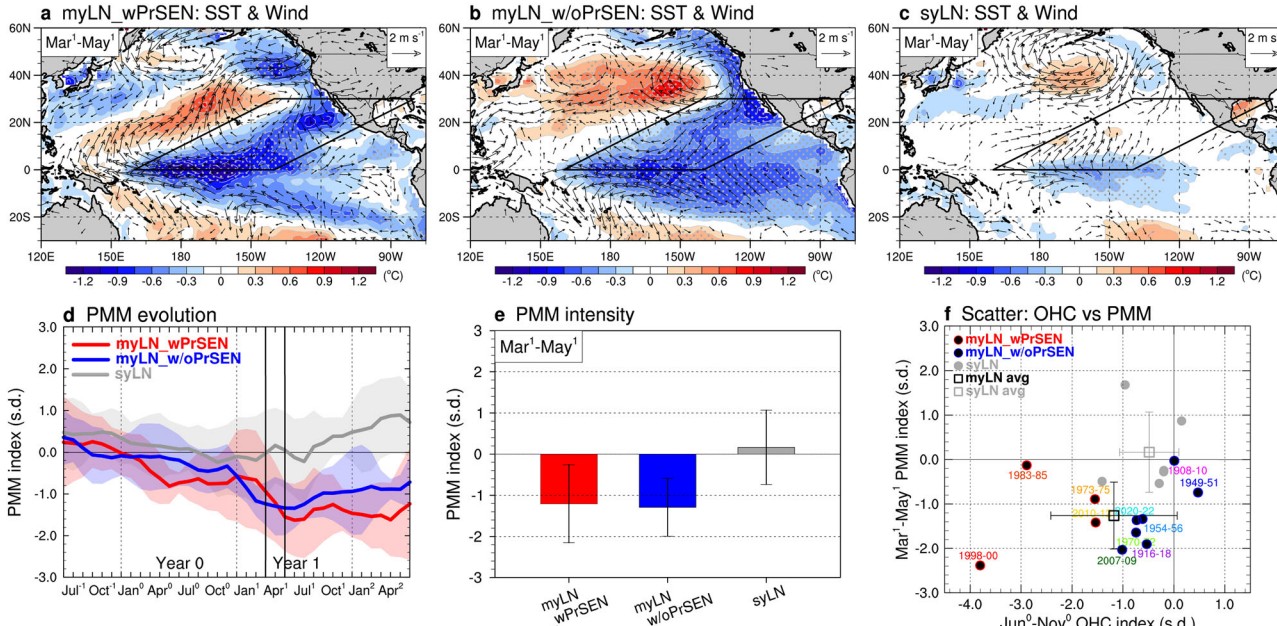

**Fig. 2 | Role of negative North Pacific Meridional Mode (PMM) in generating multi-year La Niña (LN). a–c** Composite structures of anomalous sea surface temperature (SST; shading, in °C) and surface wind (vector with minimum intensity > 0.4 m s⁻¹) during the second spring for the three LN groups: myLN_wPrSEN (**a**), myLN_w/oPrSEN (**b**), and syLN (**c**). The black parallelograms delineate the region where a negative PMM occurs and exerts its influence. The gray stippled areas indicate a significance level of 0.05 as determined by a two-tailed Student's *t*-test. **d** Evolution of PMM index for the three LN groups, with shadings representing their ±1 s.d. The index is smoothed with a 3-month running-mean filter. **e** Intensities of the second spring PMM index for the three LN groups, with error bars indicating their ±1 s.d. **f** Same as Fig. 1b, except using the ocean heat content (OHC) index values during the first fall (x-axis) plotted against the PMM index during the second spring (y-axis).

January¹ to April¹. The southern flank of this anticyclone is positioned east of Hawaii within the PMM region (Supplementary Fig. 4a, b). Note that a lagged correlation map of sea level pressure and wind anomalies during January¹ to April¹ against the precipitation index during December⁰ to February¹ (TCP-Pr index; see 'Definition of climate indices' in Methods) further solidifies the causality between the suppression and the subtropical anticyclone anomaly (Supplementary Fig. 5). In turn, this subtropical anticyclone anomaly initiates the positive wind–evaporation–SST (WES) feedback (a thermodynamic air-sea coupling between wind-induced latent heat fluxes and the underlying SSTs)[48] in the PMM region, leading to the occurrence of a negative PMM during the second spring (Fig. 2a, b and Supplementary Fig. 3b).

The interplay between cold SST and northeasterly anomalies in the subtropical northeastern Pacific (15°N–25°N, 150°W–120°W), which commences at the beginning of the second year and continues thereafter, bolsters these processes (Fig. 3d, e). The negative PMM during the second spring can persist for multiple seasons and simultaneously spread its anomalies to the equator, providing opportunities for the formation of multi-year LN (Fig. 2a, b, d). This is because, as the negative PMM-associated cold SST and northeasterly anomalies move towards the equator, they strengthen the Pacific easterly trade winds and transport anomalously cold water into the tropical central Pacific, creating the conditions for another LN to occur (Fig. 3a, b, d, e). This PMM mechanism incorporates two-way interactions within the Pacific between tropical ENSO and subtropical PMM, exerting positive feedbacks that sustain or re-intensify ENSO conditions[9–12,17,44]. Various physical processes are linked to the activation of the PMM mechanism, including the WES feedback[42,44], trade wind charging[49], oceanic Rossby wave reflection[50], and summer deep convection response[51]. A similar mechanism, involving a positive PMM often induced by central Pacific El Niño events and prone to leading to multi-year El Niño events, has also been reported[12,17] (Supplementary Text 1 for details).

In syLN, the PMM mechanism is less probable because the first LN's cold anomalies are limited in extent and mainly located in the tropical eastern Pacific, away from the western Pacific warm pool (Fig. 3c). As a result, these anomalies produce feeble atmospheric suppression in the tropical central Pacific (Supplementary Figs. 2c and 3a), which, in turn, fails to induce the subtropical anticyclone anomaly (Supplementary Fig. 4c). There is thus no interplay between SST and wind anomalies in the subtropical northeastern Pacific (Fig. 3f), resulting in the absence of a negative PMM (Fig. 2c and Supplementary Fig. 3b). Instead, anomalous northwest Pacific cyclone and accompanying westerly anomalies over the equatorial western Pacific, caused by an LN event[52,53], trigger oceanic downwelling Kelvin waves. These waves deepen the thermocline depth or raise sea surface height, pushing the underlying warm anomalies eastward and initiating the Bjerknes feedback (a positive feedback loop that strengthens the variations between surface winds, thermocline depth, and SSTs)[54] (Fig. 3c and Supplementary Fig. 6). This feedback leads to the development of an El Niño in the second year, producing a single-year event.

The skewness of the ENSO SST anomaly, with strong LN events being located further west in the tropical central Pacific than strong El Niño events[55], may play a certain role in the occurrence of the PMM. This is supported by the relatively stronger LN amplitudes in multi-year events compared to single-year events (Fig. 1c). However, it should also be noted that this relationship is not always consistent. The cold anomalies of myLN_w/oPrSEN are observed to be zonally more extended than those of myLN_wPrSEN, with their maximum centered near 120°W in the tropical eastern Pacific (Fig. 3a, b).

It is pertinent to address the question of whether the PMM is considered independent of ENSO influence or a part of it[56]. To do so, we performed additional analyses on the SST and wind maps shown in Fig. 2a–c, with a specific focus on the 'ENSO removed space' (see 'ENSO removal' in Methods). The ENSO removal was achieved by regressing out the Niño3.4 index from SST/wind fields prior to analysis. The results are in general consistent with the original findings, as the

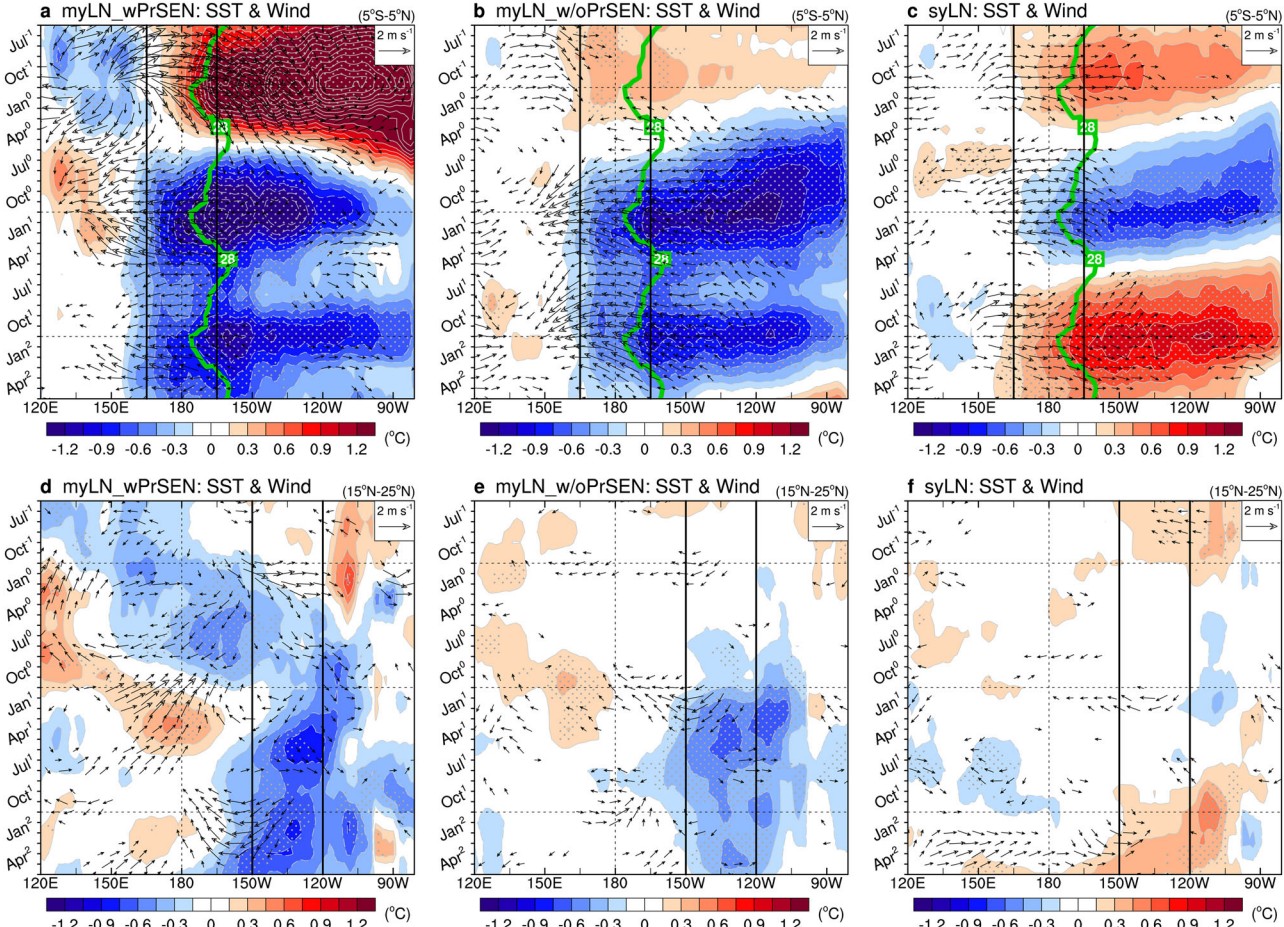

**Fig. 3 | Physical mechanism behind the occurrence of a negative North Pacific Meridional Mode (PMM). a–c** Longitude-time plots of tropical (5°S–5°N) Pacific anomalous sea surface temperature (SST; shading, in °C) and surface wind (vector with minimum intensity > 0.4 m s⁻¹) for the three La Niña (LN) groups: myLN_wPr-SEN (**a**), myLN_w/oPrSEN (**b**), and syLN (**c**). The green contours denote the climatological 28°C isotherm line at the equator, which depicts the eastern edge of the western Pacific warm pool. **d–f** Same as (**a–c**) respectively, except for subtropical (15°N–25°N) Pacific. The gray stippled areas indicate a significance level of 0.05 as determined by a two-tailed Student's *t*-test.

defining characteristics of a negative PMM are only observed in the multi-year LN groups, but not in the single-year LN group (Supplementary Fig. 7). However, it is also noticeable that the overall PMM intensities are fairly weakened in the ENSO removed space when compared to the original results (Fig. 2a, b and Supplementary Fig. 7a, b). Table 2, displaying the averaged PMM index values with ENSO removal for the myLN_wPrSEN and myLN_w/oPrSEN events, quantifies this weakening by showing that the PMM intensities are ~25% lower compared to those with no ENSO removal (cf. values in the 2ⁿᵈ and 3ʳᵈ rows). Based on these findings, we conclude that while the PMM is partially independent of ENSO, it is also dependent on ENSO to some extent and thus influenced by it.

The changes in PMM intensities with ENSO removal using various Niño indices, which also are presented in Table 2, further support our conclusion. Specifically, the weakening of PMM intensities is more pronounced when conducting ENSO removal using the Niño4 index (SST anomaly averaged in the tropical central Pacific) compared to using the Niño3.4 index (SST anomaly averaged in the tropical central-to-eastern Pacific). In contrast, when using the Niño3 or Niño1 + 2 index (SST anomaly averaged in the tropical eastern or far-eastern Pacific, respectively), the reduction rates of PMM intensities are smaller than those using the Niño3.4 index. These results imply that when an LN event extends westward into the tropical central Pacific, rather than eastward into the tropical eastern or far-eastern Pacific, its influence on the negative PMM becomes stronger.

## Application for triple-dip La Niña

To explore the potential importance of the PMM mechanism in generating triple-dip LN events, we further examined the 11 multi-year LN events and found that five of them were triple-dip events (Table 1). Of these, only two events in 1973–1975 and 1998–2000 were preceded by a strong El Niño, while the others in 1908–1910, 1954–1956, and 2020–2022 were not. This supports our argument that the role of preceding strong El Niño in the formation of multi-year LN events via the OHC mechanism has been overemphasized.

Our analysis of the Niño3.4 and PMM indices, as shown by the blue and black curves in Fig. 4a, suggests that the PMM mechanism also plays a crucial role in generating triple-dip LN events. Throughout the lifetime of a triple-dip LN, the PMM mechanism is continuously activated during both the second and third springs, where a strong negative PMM occurs and maintains its intensity. The cold SST and northeasterly anomalies over the subtropical North Pacific during these two springs confirm the occurrence of a negative PMM (Fig. 4a–c, f), which leads to the activation of the PMM mechanism in two consecutive years, completing a triple-dip event (Fig. 4e). As previously explained, the cold anomalies during the mature stage of the second LN facilitate the occurrence of a negative PMM in the third spring by extending westward across the eastern edge of the warm pool (Fig. 4e, f). However, after the third LN peaks, the cold anomalies dissipate rapidly, resulting in the absence of a strong negative PMM in the fourth spring (Fig. 4d–f). This absence prevents the activation of

**Table 2 | North Pacific Meridional Mode (PMM) intensity with the El Niño/Southern Oscillation (ENSO) signal removed**

| PMM index during March[1]–May[1] | myLN_wPrSEN | myLN_w/oPrSEN |
|---|---|---|
| No ENSO removal | −1.21 | −1.29 |
| ENSO removal using Niño3.4 index | −0.89 (−26%) | −0.99 (−23%) |
| ENSO removal using Niño4 index | −0.48 (−60%) | −0.72 (−44%) |
| ENSO removal using Niño3 index | −1.10 (−9%) | −1.09 (−16%) |
| ENSO removal using Niño1 + 2 index | −1.20 (−1%) | −1.23 (−5%) |

PMM index values (in s.d.) during the second spring (March[1]–May[1]) are shown for both multi-year La Niña (LN) groups: myLN_wPrSEN and myLN_w/oPrSEN. The values are presented with no ENSO removal (2nd row) and with ENSO removal using Niño3.4, Niño4, Niño3, and Niño1 + 2 indices (3rd to 6th row, respectively) (see 'Definition of climate indices' and 'ENSO removal' in Methods). Reduction rates, expressed as percentages relative to the values in the 2nd row, are denoted in parentheses.

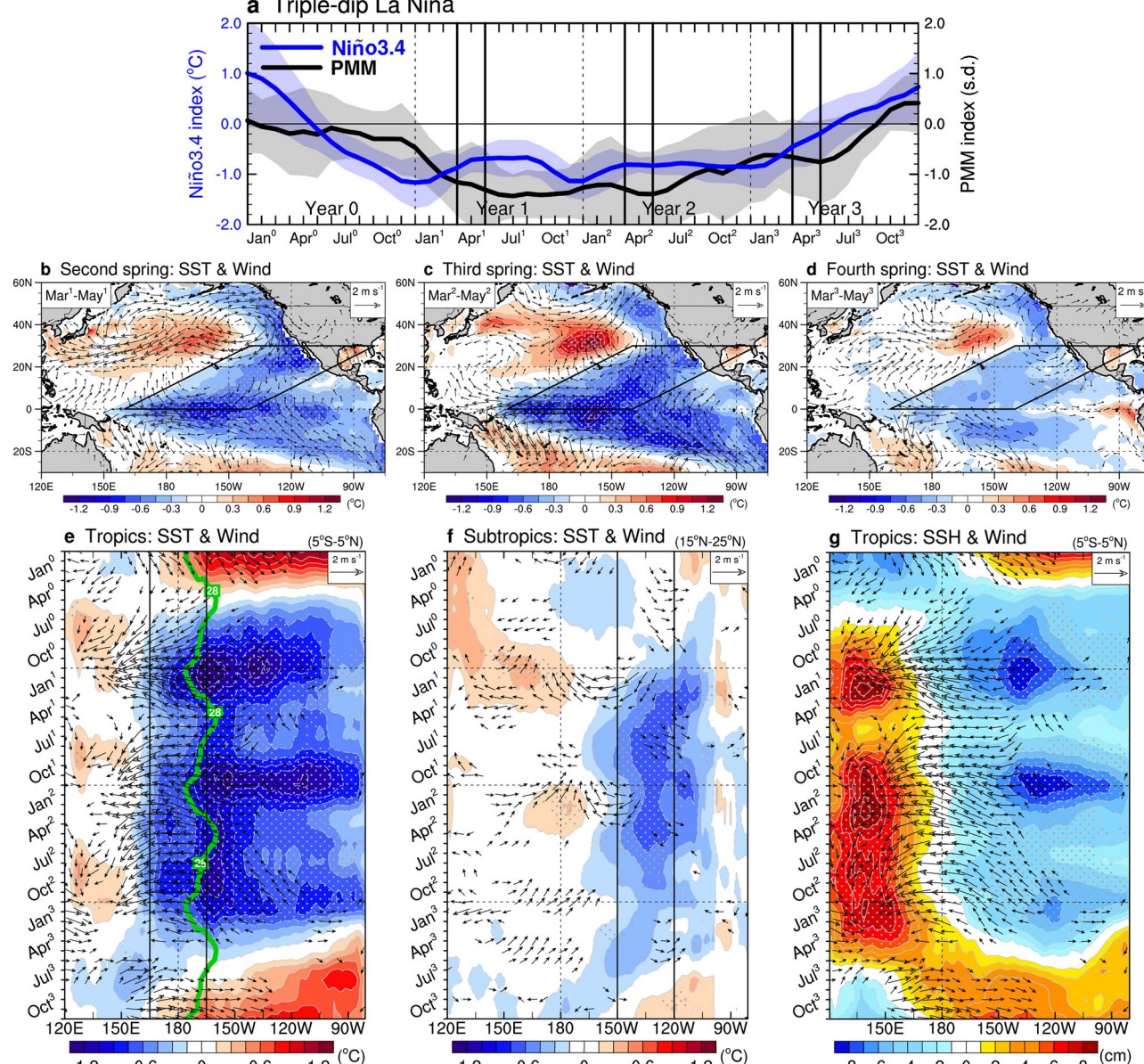

**Fig. 4 | Application of the North Pacific Meridional Mode (PMM) mechanism to the formation of triple-dip La Niña (LN). a** Composite evolutions of Niño3.4 (blue curve) and PMM (black curve) indices for the selected triple-dip LN events. The indices are smoothed with a 3-month running-mean filter and the shaded areas represent their ±1 s.d. **b–d** Composite structures of anomalous sea surface temperature (SST; shading, in °C) and surface wind (vector with minimum intensity > 0.4 m s⁻¹) for the triple-dip LN during the second (**b**), third (**c**), and fourth springs (**d**). The black parallelograms delineate the region where a negative PMM occurs and exerts its influence. **e**, **f** Longitude-time plots of anomalous SST and surface wind for the triple-dip LN over the tropical (5°S–5°N) (**e**) and subtropical (15°N–25°N) (**f**) Pacific basins. The green contour in **e** denotes the climatological 28°C isotherm line at the equator, which depicts the eastern edge of the western Pacific warm pool. **g** Same as (**e**) except for anomalous sea surface height (SSH; shading, in cm). The gray stippled areas indicate a significance level of 0.05 as determined by a two-tailed Student's *t*-test.

the PMM mechanism to produce a fourth LN. Instead, the phase subsequently transitions into an El Niño in the fourth year. The phase transition processes are similar to those observed in syLN (Fig. 3c and Supplementary Fig. 6), involving the LN-caused westerly wind forcing over the equatorial western Pacific, excitement of oceanic downwelling Kelvin waves that push the warm anomalies towards the east, and initiation of the Bjerknes feedback that elicits El Niño development (Fig. 4e, g).

## Discussion

In this study, we revisited the conventional mechanism for the formation of multi-year LN events, linking them with preceding strong El Niño. Our findings demonstrated that the majority (64%) of the observed multi-year LN events during 1900–2022 did not require a preceding strong El Niño, suggesting that the significance of this mechanism has been overemphasized. Instead, we proposed an alternative mechanism associated with a negative PMM in the subtropical North Pacific, referred to as the PMM mechanism, which plays a crucial role in generating these events. The PMM mechanism serves as a mediator or an in-between mechanism that enables an LN, if it occurs in an optimal position, to continually trigger another LN in the second or even third year, giving rise to a multi-year LN event. These findings not only contribute to advancing our understanding of ENSO dynamics but also hold substantial socio-economic relevance, given the intense and prolonged climate impacts associated with multi-year LN events.

The robustness of our findings may be affected by the limited sample sizes in observations. This limitation is inevitable for all observational studies investigating the complex behaviors of ENSO, and it cannot be resolved until a substantial number of events are accumulated over the next few decades[57]. To address the sampling issue, we conducted further analyses using a 2200-year-long model simulation by the Community Earth System Model, version 1 (CESM1)[58]. The CESM1 has been proven to accurately reproduce the observed complex behaviors of ENSO, including diverse evolution patterns such as single- and multi-year ENSO events[16,17,31,35]. It is noteworthy that CESM1 simulation has been exclusively employed and discussed in the research by Kim and Yu[17], emphasizing the controlling role of pantropical climate interactions (involving inter-basin interactions between the Pacific, Indian, and Atlantic Oceans and intra-basin interactions within the tropical and subtropical Pacific Oceans) in generating single- and multi-year ENSO events. As shown in Supplementary Table 2, the CESM1 simulation identified 182 multi-year and 49 single-year LN events, which are 17 and 8 times larger than those from the observations, respectively. Out of the 182 multi-year LN events, 71 events (39%) were classified as the myLN_wPrSEN group, while the remaining 111 events (61%) were classified as the myLN_w/oPrSEN group. This indicates that the majority of multi-year LN events in the simulation were not preceded by a strong El Niño, which is consistent with the observations. Moreover, despite some minor discrepancies, the simulation results shown in Supplementary Figs. 8–11, which were obtained by reproducing the same figures as Figs. 1–4, strongly support the findings of observational analyses (Supplementary Text 2 for details). Thus, even though it is based on a single model simulation, it can be concluded that the CESM1 simulation corroborates the robustness of the observational findings. Further study encompassing multiple climate model simulations, such as those participating in the Coupled Model Intercomparison Project Phase 6 (CMIP6)[59], will be necessary to draw more comprehensive conclusions.

It is worth discussing whether multi-year LN events, compared to single-year ones, play a critical role in generating a strong El Niño in the year following their decay. This discussion is relevant to the latest 2020–2022 triple-dip LN event, as it potentially leads to the emergence of a strong 2023/24 El Niño[60]. According to Table 1, we found that 4 out of 6 double-dip LN events and 3 out of 5 triple-dip LN events were

followed by El Niño conditions in subsequent winters (see values in italics). The following El Niño amplitudes, measured by Niño3.4 index values, showed a wide ranged from 0.64 °C to 1.87 °C when considering all eleven multi-year LN events. The average El Niño amplitude following these events is $1.23 \pm 0.43$ °C, which is not significantly larger than that following all six single-year LN events, with an average amplitude of $0.94 \pm 0.36$ °C. In view of this, our analyses do not provide evidence supporting the idea that multi-year LN events might be crucial for the formation of a strong El Niño in the subsequent year.

Lastly, the findings of this study also have implications in the context of climate change. Liguori and Lorenzo[61] reported a robust strengthening of the projected PMM variability due to anthropogenic forcing. This strengthening is expected to more easily activate the PMM mechanism, raising the possibility of an increased frequency of multi-year LN events under global warming. This possibility finds support in a recent study by Geng et al. [41], using all available emission scenarios from the CMIP6 model simulations, which revealed a significantly higher occurrence of multi-year LN events in the 21st century relative to the 20th century. Hence, in a changing climate, the significance of the negative PMM and its associated mechanism in shaping multi-year LN events could be further highlighted.

## Methods
### Observational data
We analyzed a number of monthly mean observational/reanalysis datasets covering the period from January 1900 to December 2022. For monthly mean sea surface temperature (SST), we used the Hadley Centre Sea Ice and Sea Surface Temperature version 1.1 (HadISSTv1.1) data[62], spanning from January 1871 to the present. The monthly mean atmospheric variables of surface winds, precipitation, and sea level pressure were obtained from two datasets: the National Oceanic and Atmospheric Administration 20th Century Reanalysis version 2 (NOAA 20CRv2) data[63] from January 1900 to December 1947 (originally covering from January 1871 to December 2012) and the National Center for Environmental Prediction/National Center for Atmospheric Research Reanalysis 1 (NCEP/NCAR R1) data[64] from January 1948 to December 2022. Similarly, the monthly mean sea surface height (SSH) was obtained from two datasets: the Simple Ocean Data Assimilation version 2.2.4 (SODAv2.2.4) data[65] from 1900 to 1979 (originally covering from January 1871 to August 2010) and the Global Ocean Data Assimilation System (GODAS) data[66] from January 1980 to December 2022.

### Identifying La Niña events and their classification
We identified 22 LN events in the observations from 1900 to 2022 using the following definition: An LN event was defined as occurring when the Niño3.4 index for the first winter (November⁰–January¹) is < −0.5 °C (or −0.54 s.d.). Calendar months during the first year of Year 0, when an LN event develops, were denoted as months⁰. Similarly, calendar months during the subsequent (preceding) years of Year 1(−1), 2(−2), 3(−3), …were denoted as months¹⁽⁻¹⁾, ²⁽⁻²⁾, ³⁽⁻³⁾, …, respectively. We then classified the LN events into three categories based on their evolution patterns: multi-year, single-year, or neither. A multi-year LN was determined as an event in which the Niño3.4 index for the second winter was also <−0.5 °C, while a single-year LN was defined as an event in which the second winter Niño3.4 index was >0.5 °C. Events that did not fit into either of these two categories were classified as "neither". To examine the role of preceding strong El Niño in the formation of multi-year LN events, we further divided the multi-year LN events into two sub-categories: these with a preceding strong El Niño (myLN_wPrSEN) and those without (myLN_w/oPrSEN). A myLN_wPrSEN event was identified as a multi-year LN in which the preceding winter Niño3.4 index was >1.5 °C. For a preceding winter Niño3.4 index value of <1.5 °C, the multi-year LN was identified as a myLN_w/oPrSEN event. Since there were no multi-year LN events with their preceding El Niño amplitudes between 1.5°C and 0.9°C (see

Table 1), the conclusions of this study remain unchanged if we relaxed the threshold temperature for a strong El Niño to 0.9°C. However, further relaxing the threshold, for instance, to 0.5°C, would largely diminish the conclusions of this study by including additional multi-year LN events, such as the 1970–1972, 2007–2009, and 2020–2022 events, that were not generated by the OHC mechanism. Finally, to investigate whether the PMM mechanism can prompt the generation of a third-year LN, we also divided the identified multi-year LN events into two additional sub-categories: triple-dip LN and double-dip LN. A triple-dip LN was defined as an event in which the third winter Niño3.4 index was <−0.5°C, while any other event was labeled as a double-dip LN. We acknowledge that concerns exist regarding the reliability of the selected LN events prior to the 1950s due to the unreliable climate observations and data assimilation techniques during this period. Therefore, careful interpretation is necessary for the LN events before the 1950s, and further study will thoroughly investigate these events for validation. One potential approach could involve utilizing proxy data, such as modern coral records or sediment cores, for comparison[67].

## Definition of climate indices

This study mainly utilized a number of climate indices based on SST, SSH, and precipitation anomalies. Anomalies were computed as anomalies obtained by subtracting the climatological seasonal cycle values for a base period of 1900–2022. The Niño3.4 index is commonly used to identify El Niño and LN events by quantifying their temporal variability and defined as the SST anomaly averaged in the tropical central-to-eastern Pacific (5°S–5°N, 170°W–120°W). Other Niño indices, such as the Niño4 (SST anomaly averaged in the tropical central Pacific; 5°S–5°N, 160°E–150°W), Niño3 (SST anomaly averaged in the tropical eastern Pacific; 5°S–5°N, 150°W –90°W), and Niño1 + 2 (SST anomaly averaged in the tropical far-eastern Pacific; 10°S–0°, 90°W–80°W) indices, were also utilized for the ENSO removal analyses. The OHC index represents alternating stages of upper oceanic heat content building-up (recharge) and discharge in the equatorial Pacific[34] and is defined as the normalized SSH anomaly averaged in the entire tropical Pacific (5°S–5°N, 120°E–80°W). The OHC index used in this study is highly correlated with other alternative indices such as the warm water volume (WWV; $r = 0.899$ over 1980–2022) and the depth averaged temperature in the upper 300 meters (T300; $r = 0.905$ over 1980–2022) (Supplementary Fig. 12a). The PMM index represents the interannual variability of coupled SST–surface wind pattern over the subtropical North Pacific[38]. It is defined as the normalized SST anomaly averaged in the subtropical northeastern Pacific (15°N–25°N, 150°W–120°W), following the suggestion by Richter et al.[56] The PMM index used in this study, with its simple computation, essentially replicates the results of the original indices of the PMM-SST ($r = 0.748$ over 1948–2022) and PMM-raw SST ($r = 0.877$ over 1948–2022) indices by Chiang and Vimont[38] (Supplementary Fig. 12b). The TCP-SST index represents the intensity of anomalous SST near the eastern edge of the western Pacific warm pool. It is defined as the raw SST averaged within the tropical central Pacific (5°S–5°N, 165°E–165°W), with no normalization applied. The TCP-Pr index is defined similarly to the TCP-SST index, except it employs the normalized precipitation anomaly instead.

## ENSO removal

A method to remove ENSO signals from SST and surface wind variables was applied to examine whether the PMM is independent of ENSO influence. The ENSO removed space (i.e., ENSO removal) was achieved by regressing out the various Niño indices prior to analysis. The formula for the ENSO removal is as follows:

$$\mathbf{X}_{rmENSO}(x, y, t) = \mathbf{X}(x, y, t) - \boldsymbol{\beta}(x, y) \times I(t) \tag{1}$$

where $\mathbf{X}_{rmENSO}$ and $\mathbf{X}$ represent the variable with the ENSO signal removed and the original variable, respectively, while $\boldsymbol{\beta}$ represents the linear regression coefficient estimated between the original variable ($\mathbf{X}$) and the Niño index ($I$). This approach follows the methodology employed in many prior studies[38,43,68–70].

## Statistical significance test

The two-tailed Student's $t$-test was performed to determine the statistical significance of the composite anomaly values in this study. In all analyses, a significance level of 0.05 (i.e., 95% confidence level) was used for the significance test. The formula for the two-tailed Student's $t$-test is as follows:

$$t = \frac{(\bar{X_1} - \bar{X_2}) - (\mu_1 - \mu_2)}{\sqrt{\frac{s_1^2}{n_1} + \frac{s_2^2}{n_2}}} \tag{2}$$

where $\bar{X_1}$ and $\bar{X_2}$ are the means of the first and second samples; $\mu_1$ and $\mu_2$ are the means of the first and second populations; $s_1$ and $s_2$ are the standard deviations of the first and second samples; $n_1$ and $n_2$ are the sample sizes of the first and second samples. A standard deviation (s.d.) was also calculated to determine the statistical robustness of the changes observed in the climate indices used in this study.

## Data availability

All data generated during the study are publicly available. The HadISSTv1.1 data are available from https://www.metoffice.gov.uk/hadobs/hadisst/. The NOAA 20CRv2 data are available from https://psl.noaa.gov/data/gridded/data.20thC_ReanV2.html. The NCEP/NCAR R1 data are available from https://psl.noaa.gov/data/gridded/data.ncep.reanalysis.html. The SODAv2.2.4 data are available from https://iridl.ldeo.columbia.edu/SOURCES/.CARTON-GIESE/.SODA/.v2p2p4/?SetLanguage=en. The GODAS data are available from https://psl.noaa.gov/data/gridded/data.godas.html. The CESM1 simulation outputs are available from https://www.cesm.ucar.edu/community-projects/lens/data-sets.

## Code availability

The codes used to generate all the main figures in this study have been archived in the Zenodo database under the accession code https://zenodo.org/record/8221820, or they can be obtained upon request from the corresponding author, J.-W.K. The codes were developed using the NCAR Command Language (NCL; https://www.ncl.ucar.edu/), which is a public access software.

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

## Acknowledgements

The research was initiated at the University of California, Irvine, under a grant from the National Science Foundation (NSF)'s Climate and Large-Scale Dynamics Program (AGS-2109539) and was subsequently carried out at the Jet Propulsion Laboratory, California Institute of Technology, under a contract with the National Aeronautics and Space Administration (NASA) (80NM0018D0004). The first author J.-W.K. was supported by the NASA Postdoctoral Program fellowship administered by the Oak Ridge Associated Universities (ORAU) under a contract with NASA. The second author J.-Y.Y. was supported by the NSF's Climate and Large-Scale Dynamics Program grant AGS-2109539. The third author B.T. was supported by the NASA Precipitation Measurement Mission Science Team (PMMST) program administered by Dr. Will McCarthy. © 2023. All rights reserved. We express our gratitude to the providers of observational data and CESM1 simulation outputs, whose contributions have made this study possible.

## Author contributions

J.-W.K and J.-Y.Y. designed the study. J.-W.K. performed the analysis, generated figures and tables, and wrote the initial manuscript. J.-W.K, J.-Y.Y., and B.T contributed to interpreting the results and revising the manuscript.

## Competing interests

The authors declare no competing interests.
