## [Peer Review File · Nature Communications]

REVIEWER COMMENTS

Reviewer #1 (Remarks to the Author):

Reviewer's comments:

This paper investigates the potential mechanisms that can lead to multi-year LNs based on the observations. The traditional discharge-recharge mechanism has been revisited, and a mechanism related to the negative PMM has been proposed that is crucially important for the occurrences of the multi-year LNs. The research topic is interesting and fundamentally important for the ENSO community to untangle the complex characteristics of ENSO behaviors. However, while the research is certainly valuable, there are several important questions that remain unanswered, which may reduce the scientific value of the present work. Further analyses are needed to verify the robustness of the results and improve the dynamical interpretations (causality) before the paper can be accepted for publication in Nature Communications.

General comments:

1. The major limitation of this work is its focus exclusively on observations, which only contain 11 multi-year and 6 single-year LNs. As a result, the interpretation of the results may be affected by sampling issues. To ensure the robustness of the findings, it would be beneficial to include analyses on CMIP6 model simulations.
2. Another fundamental question is unclear here is that if the PMM is but part of the ENSO evolution. Are ENSO and the PMM statistically and dynamically distinguishable? Or if the PMM is but the expression of the ENSO recharging phase? For example, the SST anomalies in the tropical Pacific region are skewed (e.g., Fig. 7 in Monahan et al. (2009)), which leads to 1) the strong LNs being located further west in the central tropical Pacific (Figs. 2a and b in this work) than strong ENs, and 2) ENs having stronger amplitudes while LNs having longer durations. Since this study found that the negative PMM tends to occur during multi-year LNs, it raises up the question if the negative PMM is nothing but the onset/decay phases of multi-year LNs.
3. The discussion of the PMM structure has primarily focused on the second/third year, making it challenging to determine whether the PMM is the cause or the result of multi-year LNs. It is lack of evidence to suggest that the PMM is more important in dynamically "generating" multi-year LN events as the authors stated in the text. The results only suggest that the negative PMM might sustain/maintain the multi-year LNs.

4. It would be worth investigating if multi-year LNs are crucial to generate (strong) ENs as well, as this could provide additional insights into studies of ENSO asymmetry.

5. While the study discusses the role of negative PMM, it would also be valuable to explore the role of positive PMM. As previous studies (e.g., Capotondi & Sardeshmukh, 2015; Capotondi et al., 2015) pointed out the PMM can be the precursor of different types of ENSO. It is therefore important to investigate whether positive PMM is crucial to generate ENs as well, which is important for understanding the ENSO diversity and asymmetry.

6. Many studies (Capotondi & Sardeshmukh, 2015; Liguori & Di Lorenzo, 2019; Min et al., 2017) pointed out the different roles of South Pacific meridional mode (SPMM) and (North) PMM in contributing to the tropical Pacific variability related to ENSO. Just wondering if there is any reason to only focus on the NPMM in this work?

Specific comments:

L42-43: Justify “up to 70%”.

L43-44: “There has also been an increasing number of LN events that persist for multiple years since the 1990s”. It is not obvious that multi-year LNs have been increased since 1990s given your Table 1, where 4 multi-year LNs were found out of 7 LN events post-1990. It might be worthwhile to justify and rephrase the sentence.

Fig. S1: It is intriguing to note that all LNs in the three categories appear to be following ENs in the preceding year, as suggested by the black curves. This result raises a follow-up question: does it suggest that the OHC-related discharge-recharge mechanism is still valid in general to explain EN->LN phase changes? Furthermore, given that this study uses a strict condition of 1.5°C to define strong ENs, it is worth asking how many strong ENs there were in total during the period of 1900-2022. This concern relates to potential sampling issues. What if the threshold for defining ENs is relaxed to 0.5°C? Would the conclusions still be valid?

Fig. 1a: very busy plot. The valuable information is buried.

L177-207: SST skewness needs to be mentioned. Also, recall general comment#3, the causality needs to be carefully revisited.

“Application for triple-dip LN” section: Sampling issues may arise due to the limited number of events selected (only five).

References:

Capotondi, A., & Sardeshmukh, P. D. (2015). Optimal precursors of different types of ENSO events: Optimal Precursors of ENSO Events. *Geophysical Research Letters*, 42(22), 9952–9960. <https://doi.org/10.1002/2015GL066171>

Capotondi, A., Wittenberg, A. T., Newman, M., Di Lorenzo, E., Yu, J.-Y., Braconnot, P., Cole, J., Dewitte, B., Giese, B., Guilyardi, E., Jin, F.-F., Karnauskas, K., Kirtman, B., Lee, T., Schneider, N., Xue, Y., & Yeh, S.-W. (2015). Understanding ENSO Diversity. *Bulletin of the American Meteorological Society*, 96(6), 921–938. <https://doi.org/10.1175/BAMS-D-13-00117.1>

Liguori, G., & Di Lorenzo, E. (2019). Separating the North and South Pacific Meridional Modes Contributions to ENSO and Tropical Decadal Variability. *Geophysical Research Letters*, 46(2), 906–915. <https://doi.org/10.1029/2018GL080320>

Min, Q., Su, J., & Zhang, R. (2017). Impact of the South and North Pacific Meridional Modes on the El Niño–Southern Oscillation: Observational Analysis and Comparison. *Journal of Climate*, 30(5), 1705–1720. <https://doi.org/10.1175/JCLI-D-16-0063.1>

Monahan, A. H., Fyfe, J. C., Ambaum, M. H. P., Stephenson, D. B., & North, G. R. (2009). Empirical Orthogonal Functions: The Medium is the Message. *Journal of Climate*, 22(24), 6501–6514. <https://doi.org/10.1175/2009JCLI3062.1>

Review of "Is a preceding strong El Niño necessary to generate multi-year La Niña events?" by Ji-Won et al.

June 2023

The manuscript questions a widely accepted hypothesis that strong El Niño (EL) events act as a precursor of multi-year(my) La Niña(LN) events. The question addressed by the authors is not only fundamental for our understanding of ENSO dynamics but has an enormous socio-economical relevance related with the numerous impacts associated with myLN events. After presenting some statistical evidence that myLN are not usually preceded by strong ENSO events, the authors propose a hypothesis for an alternative myLN precursor that involve another known interannual mode of variability of the region, the North Pacific Meridional Mode (PMM). While the author presents a simple but compelling statistical analysis that the role of strong ENSO as precursor of myLN events might have been overemphasized, they did not fully convince me that negative PMM is an actual myLN precursor. They need to provide evidence for the chain of events that might result in feedback between the westward extension of the negative SST anomalies associated with LN and the triggering of a negative PMM event.

The manuscript, which is clear, well-written and with convincing arguments supported by simple but well-posed analysis, improve our understanding regarding myLN events and more broadly of ENSO dynamics. However, I have two main comments and few suggestions I would like to the authors to address before the manuscript will be suitable for publication with Nature Communications.

Major comments:

PMM and ENSO removal: The extract the variability associated with the PMM, an index tracking ENSO variability is generally regressed out. In the original work by Chiang and Vimont (2004) the PMM is obtained by linearly removing (regressing out) the Cold Tongue Index, and nowadays a common choice is to remove the Nino3.4 index. This step is crucial as the dominance of ENSO variability in tropical Pacific SSTs would covers this partially ENSO-independent variability in the North subtropical Pacific that we refer to as PMM. I would like to see the SST/Wind maps shown in the main text also in

this “ENSO removed space” as it is crucial to distinguish between ENSO decay dynamics and actual PMM variability.

LN-PMM-LN Dynamics: Chain of events

Perhaps the main contribution of this study is the proposed chain of events that results in feedback between LN and PMM. The authors put forward an interesting hypothesis for this LN-PMM connection, that is explained in the text with the following lines:

These westward-extended cold anomalies can cause negative diabatic heating anomalies in the tropical central Pacific (165°E to 165°W) by readily reducing SSTs below the convective threshold temperature⁴², which suppresses deep convection in that region. This suppression excites stationary atmospheric Rossby waves into the North Pacific, inducing a subtropical anticyclone anomaly near Hawaii within the PMM region.

However, despite the fundamental role played by these dynamics in supporting author’s claims, this chain of events is not proved but left as speculations. I think the authors need to prove what they wrote in the above lines by performing an additional set of analysis focussing on diabatic heating, convective activity, Rossby waves propagation, and sea-level anomalies. Also, looking at the cross-correlation function might help pinning down causality relationships.

Minor comment:

The authors refer to PMM but they are actually analysing the NPMM. I agree that there is confusion in the literature as the original definition of PMM has somewhat implicitly evolved in NPMM after that Zhang et al., 2014 defined the South PMM (SPMM). I suggest replacing PMM in NPMM or at least informing the reader that PMM is known in literature also as NPMM.

Line 247: Given the reduced number of myLN event in the records, the authors use data from 1900 to 2022, which contains highly unreliable decades (~1900-1950). The validity of the results for the first half of the 20th century should be discussed in the main text with a couple of sentences.

Line 262 and 277: expressed the threshold -0.5°C also in units of STD

Line 330: Codes should be available without request to the corresponding author. They can use zenodo.

I think might be worth discussing the implications of authors' findings in the context of climate change and projected changes in PMM variability. Specifically, under anthropogenic forcing one might expect an increase in the variance of myLN events because of the robust projected increase in the variance of the PMM (Liguori and Di Lorenzo, 2018), which the authors' claim to be a precursor of myLN.

Liguori G. and Di Lorenzo E., 2018: Meridional Modes and Increasing Pacific decadal variability under greenhouse forcing. *Geophysical Research Letters*, 45. <https://doi.org/10.1002/2017GL076548>.

Point-by-Point Responses to Reviewer #1

This paper investigates the potential mechanisms that can lead to multi-year LNs based on the observations. The traditional discharge-recharge mechanism has been revisited, and a mechanism related to the negative PMM has been proposed that is crucially important for the occurrences of the multi-year LNs. The research topic is interesting and fundamentally important for the ENSO community to untangle the complex characteristics of ENSO behaviors. However, while the research is certainly valuable, there are several important questions that remain unanswered, which may reduce the scientific value of the present work. Further analyses are needed to verify the robustness of the results and improve the dynamical interpretations (causality) before the paper can be accepted for publication in Nature Communications.

Response: We would like to express our gratitude to the reviewer for dedicating time to review our manuscript. We greatly value the insightful comments and suggestions provided by the reviewer, which have significantly improved our manuscript. In response to the reviewer's comments, we have conducted further analyses addressing various aspects, including comment 1 (the limited sample size in observations), comment 2 (the independence of the PMM from ENSO), comment 4 (the importance of multi-year La Niña events in generating strong El Niño events), comment 8 (the increase in the number of multi-year La Niña events since the 1990s), comment 9 (the threshold for a preceding strong El Niño), and comment 10 (the clarity of Fig. 1a). We have also appropriately incorporated all other comments and suggestions raised by the reviewer into the revised manuscript. Our detailed point-by-point responses to the reviewer are as follows.

General comments:

1. The major limitation of this work is its focus exclusively on observations, which only contain 11 multi-year and 6 single-year LNs. As a result, the interpretation of the results may be affected by sampling issues. To ensure the robustness of the findings, it would be beneficial to include analyses on CMIP6 model simulations.

Response: We thank the reviewer for the constructive comment. We acknowledge the concern regarding the limited sample sizes of multi-year and single-year La Niña events in this study. It is however important to note that all studies investigating the complex behaviors of ENSO encounter the challenge of a short observational record, which inevitably leads to the issue of small sample sizes. Unfortunately, this issue cannot be fully resolved until we

have a substantial number of real-world events accumulated over the course of several decades (e.g., Kim et al., 2023).

To address the sampling issue resulting from limited observations, we conducted a new analysis using a 2200-year long-integrated model simulation generated by the Community Earth System Model, version 1 (CESM1; Kay et al., 2015). The CESM1 has been proved to realistically simulate the observed complex behaviors of ENSO, including diverse evolution patterns such as single- and multi-year ENSO events (e.g., DiNezio et al., 2017; Wu et al., 2019; Kim & Yu, 2022; Zhu & Yu, 2022). This model simulation data is accessible via the Earth System Grid at <https://www.cesm.ucar.edu/community-projects/lens/data-sets>. In the CESM1 analysis, we identified 182 multi-year and 49 single-year La Niña events. These events were approximately 17 times larger for multi-year events and 8 times larger for single-year events compared to those observed (Table A1). As shown in Table A1, out of the 182 multi-year La Niña events in CESM1, 71 (or 39%) were classified as multi-year La Niña events with a preceding strong El Niño (also referred to as myLN_wPrecEN), and 111 (or 61%) were identified as events without (also referred to as myLN_w/oPrecEN). This indicates that the majority of multi-year LN events in the simulation were not preceded by a strong El Niño, which is consistent with the observations. The remaining 152 La Niña events that did not fit into either the multi-year or single-year category were classified as “neither” events. The methods used to identify and classify La Niña events are the same as those in the observations, except for the use of ± 0.5 standard deviations ($\pm 0.57^\circ\text{C}$) instead of $\pm 0.5^\circ\text{C}$ (refer to “Methods” section in the original manuscript).

Table A1 | Numbers of La Niña events classified in the CESM1 simulation and the observations. Numbers in parentheses in the “Multi-year La Niña” column denote the occurrences of triple-dip La Niña events.

Number of events	Multi-year La Niña	w/ a preceding strong El Niño	w/o a preceding strong El Niño	Single-year La Niña	Neither La Niña
CESM1	182 (40)	71	111	49	152
OBS	11 (5)	4	7	6	5

Using the events obtained from the aforementioned CESM1 simulation, we have reproduced Figures 1 to 4 as outlined in the manuscript. The corresponding figures are presented here as

Figures A1 to A4. In short, albeit with some minor discrepancies such as slightly more westward-extended La Niña structures, the new analyses and results from the simulation strongly support the findings of observational analyses, revealing the following:

- (i) Fig. A1: The excessively emphasized role of preceding strong El Niño in generating multi-year La Niña events (refer to Sect. “Role of preceding strong El Niño: Overemphasized” in the original manuscript)
- (ii) Fig. A2: The crucial role of negative PMM in generating multi-year La Niña events (refer to Sect. “Role of negative PMM: Crucial” in the original manuscript)
- (iii) Fig. A3: The physical mechanism responsible for the occurrence of a negative PMM (refer to Sect. “Role of negative PMM: Crucial” in the original manuscript)
- (iv) Fig. A4: The applicability of the PMM mechanism for the generation of triple-dip La Niña events (refer to Sect. “Application for triple-dip LN” in the original manuscript).

Consequently, the robustness of our findings from observations is ensured by the results from a long-term CESM1 simulation that provides a large sample size. **We have stated these new results and their related discussions in the revised manuscript (Lines 305-322). Table A1 and Figures A1-A4 have been added as Supplementary Table 2 and Supplementary Figures 8-11 in the revised Supplementary Information, respectively. Supplementary Text 1 also provides a detailed description of the CESM1 analyses and the associated figures.**

We recognize that presenting results from a single climate model simulation may not sufficiently ensure the robustness of the findings of this study. However, conducting analyses on CMIP6 model simulations, which encompass more than 30 available model simulations (Eyring et al., 2016), will require extensive new analyses from scratch. These analyses should include model performances and validations to account for the inherent deficiencies/biases of the models in simulating complex ENSO behaviors (cf. Planton et al., 2021). We believe that such an endeavor would exceed the scope of this study and should be systematically carried out in a future study to provide comprehensive and meaningful outcomes. **This point has also been addressed in the revised manuscript (Lines 322-324).**

Refs.

Kim, J. W., Chang, T. H., Lee, C. T., & Yu, J. Y. (2023). Evaluating ENSO’s Spatiotemporal Complexity in the CWB CFS 1-Tiered Model Hindcasts. *Journal of Geophysical Research*:

Atmospheres, e2022JD038200.

Kay, J. E., Deser, C., Phillips, A., Mai, A., Hannay, C., Strand, G., ... & Vertenstein, M. (2015). The Community Earth System Model (CESM) large ensemble project: A community resource for studying climate change in the presence of internal climate variability. *Bulletin of the American Meteorological Society*, 96(8), 1333-1349.

DiNezio, P. N., Deser, C., Okumura, Y., & Karspeck, A. (2017). Predictability of 2-year La Niña events in a coupled general circulation model. *Climate dynamics*, 49(11-12), 4237-4261.

Wu, X., Okumura, Y. M., & DiNezio, P. N. (2019). What Controls the Duration of El Niño and La Niña Events?. *Journal of Climate*, 32(18), 5941-5965.

Kim, J. W., & Yu, J. Y. (2022). Single-and multi-year ENSO events controlled by pantropical climate interactions. *npj Climate and Atmospheric Science*, 5(1), 88.

Zhu, T., & Yu, J. Y. (2022). A Shifting Tripolar Pattern of Antarctic Sea Ice Concentration Anomalies During Multi-Year La Niña Events. *Geophysical Research Letters*, 49(23), e2022GL101217.

Eyring, V., Bony, S., Meehl, G. A., Senior, C. A., Stevens, B., Stouffer, R. J., & Taylor, K. E. (2016). Overview of the Coupled Model Intercomparison Project Phase 6 (CMIP6) experimental design and organization. *Geoscientific Model Development*, 9(5), 1937-1958.

Planton, Y. Y., Guilyardi, E., Wittenberg, A. T., Lee, J., Gleckler, P. J., Bayr, T., ... & Voldoire, A. (2021). Evaluating climate models with the CLIVAR 2020 ENSO metrics package. *Bulletin of the American Meteorological Society*, 102(2), E193-E217.

Fig. A1 | Same as Fig. 1 in the original manuscript except using the CSM1 simulation

Fig. A2 | Same as Fig. 2 in the original manuscript except using the CESM1 simulation

Fig. A3 | Same as Fig. 3 in the original manuscript except using the CESM1 simulation

Fig. A4 | Same as Fig. 4 in the original manuscript except using the CESM1 simulation

2. Another fundamental question is unclear here is that if the PMM is but part of the ENSO evolution. Are ENSO and the PMM statistically and dynamically distinguishable? Or if the PMM is but the expression of the ENSO recharging phase? For example, the SST anomalies in the tropical Pacific region are skewed (e.g., Fig. 7 in Monahan et al. (2009)), which leads to 1) the strong LNs being located further west in the central tropical Pacific (Figs. 2a and b in this work) than strong ENs, and 2) ENs having stronger amplitudes while LNs having longer durations. Since this study found that the negative PMM tends to occur during multi-year LNs, it raises up the question if the negative PMM is nothing but the onset/decay phases of multi-year LNs.

Response: We thank the reviewer for raising these important questions. Firstly, we would like to briefly highlight that this study has identified a limitation in the traditional understanding of the preceding strong El Niño-related OHC mechanism for generating multi-year La Niña events. This limitation arises from the finding that more than half (64%) of the observed multi-year La Niña events do not necessitate a preceding strong El Niño (also referred to as myLN_w/oPrecEN events). By comparing these myLN_w/oPrecEN events with the remaining multi-year La Niña and single-year La Niña events (also referred to as myLN_wPrecEN and syLN events, respectively), this study has revealed that the OHC mechanism is not effective for the myLN_w/oPrecEN events (refer to Fig. 1 in the original manuscript). These findings have led us to the alternative mechanism, the PMM mechanism proposed in this study, for the generation of these multi-year La Niña events (refer to Figs. 2 and 3 in the original manuscript).

Regarding the first question in the comment (“**Are ENSO and the PMM statistically and dynamically distinguishable? Or if the PMM is but the expression of the ENSO recharging phase?**”), previous studies have clearly demonstrated that the generation mechanism of the PMM results from the wind-evaporation-SST feedback operating in the subtropical North Pacific and does not involve the recharge/discharge of ocean heat content in the tropical Pacific (Amaya, 2019 and references therein). The PMM can be triggered by ENSO, but it does not imply that it is simply a part of the ENSO, as evidenced by the fact that not all ENSO events can trigger the PMM. Fang and Yu (2020) have shown that only the Central Pacific type of ENSO events is more capable of triggering the PMM through the Rossby wave responses excited by the ENSO diabatic heating anomaly. Since multi-year La Niña events tend to be located in the tropical central Pacific, they are more capable of

triggering the PMM (refer to Fig. 3a, b, d, e in the original manuscript). However, as we can see from the single-year La Niña events, which are typically located in the tropical eastern Pacific, they cannot trigger the PMM (refer to Fig. 3c, f in the original manuscript). The existence of single-year La Niña events thus indicates that the PMM is not a part of the La Niña evolution in the ENSO recharging phase. The controlling factor of the PMM, namely the longitudinal location of cold anomalies during a La Niña, cannot be influenced by the ENSO recharging phase itself.

Meanwhile, as the reviewer pointed out, the skewness of ENSO SST anomaly, with strong La Niña events being located further west in the tropical central Pacific and having longer durations than strong El Niño events, may play a certain role in the occurrence of the PMM. This is supported by the relatively stronger La Niña amplitudes observed in multi-year events compared to single-year events (refer to Fig. 1c in the original manuscript). However, it should also be noted that this relationship is not always consistent. The cold anomalies of the myLN_w/oPrecEN group are observed to be zonally more extended than those of the myLN_wPrecEN group, with their maximum centered near 120°W in the tropical eastern Pacific (refer to Fig. 3a, b in the original manuscript).

Regarding the second question in the comment (“**Since this study found that the negative PMM tends to occur during multi-year LNs, it raises up the question if the negative PMM is nothing but the onset/decay phases of multi-year LNs.**”), we would like to emphasize that more than half of the observed multi-year La Niña events were not generated by the OHC mechanism, as noted earlier. If the negative PMM were indeed nothing more than a manifestation during onset/decay phases of multi-year La Niña events, it would appear that every multi-year La Niña event was solely generated by the OHC mechanism. To further address our argument, we conducted additional analyses on the SST/Wind maps presented in the main text (refer to Fig. 2a-c in the original manuscript), with a specific focus on the “ENSO removed space”. This so-called ENSO removed space, achieved by regressing out the Niño3.4 index from SST/Wind fields prior to analysis, allowed us to examine the PMM independently of ENSO influence. The results shown in Figure A5 here indicate that the defining characteristics of a negative PMM (i.e., cold SST and accompanying northeasterly anomalies in the subtropical North Pacific) are observed in both multi-year La Niña groups (Fig. A5a and A5b). In contrast, these characteristics are not observed in the single-year La

Niña group (Fig. A5c). These are overall consistent with the original results obtained with no ENSO removal.

We have incorporated our responses to the two questions and the additional analyses into the revised manuscript (Lines 244-258). Figure A5 has been added as Supplementary Figure 7 in the revised Supplementary Information. The study of Monahan et al. (2009) has been newly cited.

Refs.

Amaya, D. J. (2019). The Pacific meridional mode and ENSO: A review. *Current Climate Change Reports*, 5, 296-307.

Fang, S. W., & Yu, J. Y. (2020). A control of ENSO transition complexity by tropical Pacific mean SSTs through tropical-subtropical interaction. *Geophysical Research Letters*, 47(12), e2020GL087933.

Monahan, A. H., Fyfe, J. C., Ambaum, M. H., Stephenson, D. B., & North, G. R. (2009). Empirical orthogonal functions: The medium is the message. *Journal of Climate*, 22(24), 6501-6514.

Fig. A5 | Same as Fig. 2a-c in the original manuscript except with ENSO signals removed by regressing out the Niño3.4 index from SST and wind fields prior to analysis

3. The discussion of the PMM structure has primarily focused on the second/third year, making it challenging to determine whether the PMM is the cause or the result of multi-year LNs. It is lack of evidence to suggest that the PMM is more important in dynamically “generating” multi-year LN events as the authors stated in the text. The results only suggest that the negative PMM might sustain/maintain the multi-year LNs.

Response: With regard to the reviewer’s statement that there is a lack of evidence to suggest that the PMM is more important in dynamically generating multi-year La Niña events, we would firstly like to highlight the compelling evidence provided by this study. The results demonstrate that the (negative) PMM serves as the key factor distinguishing La Niña events evolving into multi-year or single-year events (refer to Fig. 2 and 3 in the original manuscript). On contrary, the traditional view, which emphasizes the role of the preceding strong El Niño, fails to make the same distinction (refer to Fig. 1 in the original manuscript). Secondly, our results demonstrate the importance of the PMM in creating opportunities for the formation of multi-year La Niña events (refer to Figs. 2 and 3 in the original manuscript). Regardless of how a 1st-year La Niña is produced, if it occurs in an optimal longitudinal position over the tropical central Pacific, it easily triggers the PMM during its decaying phase (refer to Figs. S2 and S3 in the original manuscript). The triggered PMM then helps generate a 2nd-year La Niña, resulting in a double-dip La Niña. If the 2nd-year La Niña also triggers the PMM during its decaying phase, then a 3rd-year La Niña arises, becoming a triple-dip La Niña (refer to Fig. 4 in the original manuscript). This is why we have primarily focused on the second and third years. In conclusion, the PMM does not just sustain/maintain the multi-year La Niña events but acts as a mediator or an in-between mechanism that enables a La Niña to trigger another La Niña, giving rise to a multi-year event. Lastly, as mentioned in our response to Comment 1 regarding the issue of a limited sample size, the findings of this study will gain further credibility from the consistency observed in the 2200-year-long CESM1 simulation, which contains a significantly larger sample size (refer to Figs. A1-A4). **In response, we have included discussions of these points into the revised manuscript (Lines 195-210, 226-228, and 296-304).**

4. It would be worth investigating if multi-year LNs are crucial to generate (strong) ENs as well, as this could provide additional insights into studies of ENSO asymmetry.

Response: We thank the reviewer for the insightful comment. We would like to note that the

reviewer's request to investigate **“if multi-year LNs are crucial to generate (strong) ENs as well”** lies beyond the scope of this study. The primary objective of this study is to enhance our understanding of the formation mechanism behind multi-year La Niña events. Nevertheless, we have made efforts to address the reviewer's request.

To investigate whether multi-year La Niña events are crucial for the generation of strong El Niño events, we checked the wintertime Niño3.4 index values following the double-dip and triple-dip La Niña events identified in this study, as shown here in Table A2. The results indicate that 4 out of 6 double-dip La Niña events were followed by El Niño conditions in the subsequent winter, with Niño3.4 index values ranging from 0.64°C to 1.87°C (values in italics). Similarly, 3 out of 5 triple-dip La Niña events were followed by El Niño conditions in the subsequent winter, with Niño3.4 index values ranging from 0.80°C to 1.26°C. Note that the data regarding potential El Niño conditions in the winter of 2023 after the 2020-22 triple-dip La Niña are not yet available. When considering all 11 multi-year La Niña events, the average amplitude of El Niño following these events is 1.23°C, with their standard deviation of $\pm 0.43^\circ\text{C}$. The results do not exhibit statistically significant distinction in El Niño amplitude compared to the results from the 6 single-year La Niña events (on average, 0.94°C with $\pm 0.36^\circ\text{C}$ standard deviation; refer to Table A2). Therefore, while further studies are still necessary, our analyses do not provide evidence supporting the idea that multi-year La Niña events are crucial for the generation of strong El Niño events in the subsequent year.

In the revised manuscript, we have incorporated these findings into the discussion section (Lines 325-335). Additionally, the original Table 1 has been modified to include the Niño3.4 index values during the fourth winter (Table 1 in the revised manuscript). The recent study by Lian et al. (2023), which proposed the possibility of the emergence of a strong 2023 El Niño after the 2020-2022 triple-dip La Niña, has been newly cited.

Ref. Lian, T., Wang, J., Chen, D., Liu, T., & Wang, D. (2023). A strong 2023/24 El Niño is staged by tropical Pacific Ocean heat content buildup. *Ocean-Land-Atmosphere Research*, 2, 0011.

Table A2 | La Niña event years classified as multi-year (double-dip or triple-dip) or single-year events in this study. The corresponding Niño3.4 index values are shown during the first, second, third, and fourth winters. Values in bold (italics) indicate La Niña (El Niño) conditions with Niño3.4 index $< -0.5^{\circ}\text{C}$ ($> 0.5^{\circ}\text{C}$).

La Niña classification		Event year	First winter Niño3.4 index	Second winter Niño3.4 index	Third winter Niño3.4 index	Fourth winter Niño3.4 index
Multi-year La Niña	Double-dip	1916/17/18	-1.59	-0.63	1.38	-
		1949/50/51	-1.13	-1.13	0.64	-
		1970/71/72	-1.10	-0.75	1.87	-
		1983/84/85	-0.85	-1.13	-0.49	-
		2007/08/09	-1.58	-0.69	1.56	-
		2010/11/12	-1.57	-0.97	-0.03	-
	Triple-dip	1908/09/10	-0.73	-1.13	-0.54	1.13
		1954/55/56	-0.84	-1.50	-0.68	1.26
		1973/74/75	-2.00	-0.69	-1.58	0.80
		1998/99/00	-1.31	-1.51	-0.78	-0.27
2020/21/22		-0.95	-0.85	-0.74	N/A	
Single-year La Niña	1903/04	-0.79	0.60	-	-	
	1924/25	-0.88	1.37	-	-	
	1938/39	-0.69	0.55	-	-	
	1964/65	-0.87	1.38	-	-	
	2005/06	-0.67	0.90	-	-	
	2017/18	-0.78	0.84	-	-	

5. While the study discusses the role of negative PMM, it would also be valuable to explore the role of positive PMM. As previous studies (e.g., Capotondi & Sardeshmukh, 2015; Capotondi et al., 2015) pointed out the PMM can be the precursor of different types of ENSO. It is therefore important to investigate whether positive PMM is crucial to generate ENs as well, which is important for understanding the ENSO diversity and asymmetry.

Response: We thank the reviewer for the insightful suggestion. However, investigating whether a positive PMM is crucial in generating multi-year El Niño events exceeds the scope of this study. Moreover, such an investigation will require additional extensive analyses and careful interpretations and discussions, especially if there are asymmetric responses between the roles of positive and negative PMM in the formation of multi-year ENSO events.

Therefore, we believe that the role of the positive PMM deserves to be addressed in a separate study.

6. Many studies (Capotondi & Sardeshmukh, 2015; Liguori & Di Lorenzo, 2019; Min et al., 2017) pointed out the different roles of South Pacific meridional mode (SPMM) and (North) PMM in contributing to the tropical Pacific variability related to ENSO. Just wondering if there is any reason to only focus on the NPMM in this work?

Response: We thank the reviewer for the careful comment. As the reviewer pointed out, many studies have not only emphasized the importance of the (North) PMM (hereafter, NPMM) but have also underscored the significance of the SPMM, a similar pattern of climate variability observed in the South Pacific (Zhang et al., 2014). Both the meridional modes are known to play some different roles in the development of ENSO events and tropical Pacific decadal variability (Capotondi & Sardeshmuck, 2015; Min et al., 2017; Liguori & Di Lorenzo, 2019). With this regard, it is widely recognized that the NPMM primarily favors the development of CP-type ENSO events, while the SPMM tends to be associated with the occurrence of Eastern Pacific (EP)-type ENSO events.

There are two reasons why this study exclusively focused on the role of the NPMM rather than the SPMM. Firstly, despite recent progress, the physical connection between the SPMM and ENSO remains unclear in the literature, without a clear consensus (cf. Amaya, 2019). In addition, there is also limited understanding of how the SPMM impacts ENSO and, if it does, how ENSO feeds back to the SPMM (refer to the “NPMM vs. SPMM” section in Amaya, 2019). Secondly, the potential role of SPMM in generating multi-year ENSO events has not been thoroughly investigated. This is likely due to the uncertainties regarding the contributions of SPMM to ENSO, particularly in relation to multi-year ENSO evolution patterns. Considering the role of SPMM would therefore raise questions that need to be carefully addressed in order to resolve the uncertainties. We believe that further studies are required to determine whether the SPMM is induced remotely from ENSO events and, if so, whether the SPMM-involved intra-basin interactions between the tropical and subtropical South Pacific Ocean contribute to the multi-year ENSO evolution patterns. **We have briefly discussed these points and incorporated them into the revised manuscript (Lines 336-343). The studies of Zhang et al. (2014), Capotondi & Sardeshmuck (2015), Min et al.**

(2017), Liguori & Di Lorenzo (2019), and Amaya (2019) have been cited as well.

Refs.

Zhang, H., Clement, A., & Di Nezio, P. (2014). The South Pacific meridional mode: A mechanism for ENSO-like variability. *Journal of Climate*, 27(2), 769-783.

Capotondi, A., & Sardeshmukh, P. D. (2015). Optimal precursors of different types of ENSO events: Optimal Precursors of ENSO Events. *Geophysical Research Letters*, 42(22), 9952–9960. <https://doi.org/10.1002/2015GL066171>

Min, Q., Su, J., & Zhang, R. (2017). Impact of the South and North Pacific Meridional Modes on the El Niño–Southern Oscillation: Observational Analysis and Comparison. *Journal of Climate*, 30(5), 1705–1720. <https://doi.org/10.1175/JCLI-D-16-0063.1>

Liguori, G., & Di Lorenzo, E. (2019). Separating the North and South Pacific Meridional Modes Contributions to ENSO and Tropical Decadal Variability. *Geophysical Research Letters*, 46(2), 906–915. <https://doi.org/10.1029/2018GL080320>

Amaya, D. J. (2019). The Pacific meridional mode and ENSO: A review. *Current Climate Change Reports*, 5, 296-307.

Specific comments:

7. L42-43: Justify “up to 70%”.

Response: Thank you for the comment. According to the recent study by Kim & Yu (2022), 70% (14 out of 20 events) of the observed La Niña events between 1900–2020 were identified as multi-year La Niña. Other previous studies have also reported percentages for multi-year La Niña, such as 43% (10 out of 23 events between 1900 and 2017) by Wu et al. (2019) and 60% (6 out of 10 events between 1961 and 2016) by Iwakiri & Watanabe (2021). It is therefore expected that the percentages can vary due to differences in the definition of multi-year La Niña events. **Based on these findings, we have revised the original writing in the revised manuscript to state “roughly more than half of LN events (43–70%)” instead of “most LN events (up to 70%)” (Lines 42-43).** The studies of Kim & Yu (2022), Wu et al. (2019), and Iwakiri & Watanabe (2021) have also been cited.

Refs.

Kim, J. W., & Yu, J. Y. (2022). Single-and multi-year ENSO events controlled by pantropical climate interactions. *npj Climate and Atmospheric Science*, 5(1), 88.

Wu, X., Okumura, Y. M., & DiNezio, P. N. (2019). What controls the duration of El Niño and La Niña events?. *Journal of Climate*, 32(18), 5941-5965.

Iwakiri, T., & Watanabe, M. (2021). Mechanisms linking multi-year La Niña with preceding strong El Niño. *Scientific reports*, 11(1), 17465.

8. L43-44: “There has also been an increasing number of LN events that persist for multiple years since the 1990s”. It is not obvious that multi-year LNs have been increased since 1990s given your Table 1, where 4 multi-year LNs were found out of 7 LN events post-1990. It might be worthwhile to justify and rephrase the sentence.

Response: Thank you for the comment. To address the concern raised by the reviewer, we have created Table A3, which presents the number of multi-year La Niña events based on our Table 1. Additionally, we have included in the table their frequency measured in events per decade during two periods: “Pre-1990” (spanning from 1900 to 1989, 90 years) and “Post-1990” (spanning from 1990 to 2022, 33 years). As demonstrated in Table A3, 7 multi-year La Niña events occurred during the Pre-1990 period, with an event frequency of 0.78 per decade. In contrast, 4 multi-year La Niña events occurred during the Post-1990 period, with an event frequency of 1.21 per decade. Consequently, the occurrence frequency of multi-year La Niña events increased by approximately 55% during the Post-1990 period compared to the Pre-1990 period. Hence, our statement, **“There has also been an increasing number of LN events that persist for multiple years since the 1990s”** is indeed accurate. **To provide further support for our statement, we have included Table A3 as Supplementary Table 1 in the revised Supplementary Information, accompanied by a detailed explanation.**

Table A3 | Statistics of the occurrence frequency of multi-year La Niña events for two distinct periods: ‘Pre-1990’ (spanning from 1900 to 1989, a total of 90 years) and ‘Post-1990’ (spanning from 1990 to 2022, a total of 33 years). The numbers of multi-year La Niña events are derived from Table 1. A notable increase of approximately 55% is observed during the Post-1990 period in comparison to the Pre-1990 period.

Period	Pre-1990 (1900–1989)	Post-1990 (1990–2022)
Number of multi-year La Niña events	7	4
Events per decade	0.78	1.21

9. Fig. S1: It is intriguing to note that all LNs in the three categories appear to be following ENs in the preceding year, as suggested by the black curves. This result raises a follow-up question: does it suggest that the OHC-related discharge-recharge mechanism is still valid in general to explain EN->LN phase changes? Furthermore, given that this study uses a strict condition of 1.5°C to define strong ENs, it is worth asking how many strong ENs there were in total during the period of 1900-2022. This concern relates to potential sampling issues. What if the threshold for defining ENs is relaxed to 0.5°C? Would the conclusions still be valid?

Response: Thank you for the comment. Regarding the first question raised by the reviewer, it is important to note that this study does not assert the invalidity of the OHC (-related discharge-recharge) mechanism in explaining the phase transition from El Niño to La Niña. The OHC mechanism remains a key process for the development of La Niña following an El Niño. The focus of this study was on the fact that the OHC mechanism does not sufficiently account for the occurrence of multi-year La Niña events. This study has provided clear evidence as more than half of the observed multi-year La Niña events (64%, 7 out of 11 events) were not preceded by a strong El Niño, which would have induced a large and persistent equatorial Pacific upper-ocean heat discharge to activate the OHC mechanism.

Regarding the reviewer’s subsequent questions about the threshold for a strong El Niño (1.5°C in this study), firstly, we note that the conclusions of this study would not change even if we relaxed the threshold to 0.9°C. That is because there were no multi-year La Niña events with preceding El Niño amplitude between 1.5°C and 0.9°C (refer to Table 1 and Fig. 1a, b in the original manuscript). Secondly, if we further relaxed the threshold to 0.5°C (allowing all

preceding El Niño events regardless of their amplitude), we would have three additional multi-year La Niña events: 1970-1972, 2007-2009, and 2020-2022, as part of the myLN_wPrecEN group in the analysis (refer to Table 1 in the original manuscript). As displayed in Fig. A6 here, the OHC mechanism, represented by the OHC index (red curves in Fig. A6), does not contribute to the generation of these three multi-year La Niña events due to the absence of any noticeable second-year OHC discharge. Therefore, including these additional events by further relaxing the threshold would diminish the conclusions of this study. **We have briefly added a statement of these points in the revised manuscript (Lines 386-391).**

Fig. A6 | Evolutions of the Niño3.4 (black curve) and OHC (red curve) indices for the multi-year La Niña events of 1970-1972 (a), 2007-2009 (b), and 2020-2022 (c).

10. Fig. 1a: very busy plot. The valuable information is buried.

Response: Thank you for the comment. **In the revised manuscript, we have replaced the original Fig. 1a with Fig. A7 shown here along with additional description (Lines 657-658).** This replacement helps reduce clutter and retain valuable information within the plot.

Fig. A7 | Evolution of the Niño3.4 index for individual multi-year La Niña events during 1900–2022. Each event year is represented by a different color, corresponding to its specific evolution.

11. L177-207: SST skewness needs to be mentioned. Also, recall general comment#2, the causality needs to be carefully revisited.

Response: Thank you for the comment. **The potential role of SST skewness has been mentioned in the revised manuscript (Lines 237-243). The causality elucidating the physical processes behind the occurrence of a negative PMM and its role in generating multi-year La Niña events has also been carefully revisited (Lines 195-210 and 226-228).** We kindly request that the reviewer also refer to our previous responses regarding Comments 2 and 3.

12. “Application for triple-dip LN” section: Sampling issues may arise due to the limited number of events selected (only five).

Response: Thank you for the comment. We acknowledge the sampling issue regarding the limited number of triple-dip La Niña events observed. To address this issue, we conducted new analyses utilizing a 2200-year-long CESM1 simulation that includes 40 triple-dip La Niña events (refer to Table A1). This number is eight times larger than those observed. As mentioned in our prior response to Comment 1, the results for the triple-dip La Niña events derived from the CESM1 simulation robustly support the observational findings (refer to Fig. A4).

Point-by-Point Responses to Reviewer #2

The manuscript questions a widely accepted hypothesis that strong El Niño (EL) events act as a precursor of multi-year (my) La Niña (LN) events. The question addressed by the authors is not only fundamental for our understanding of ENSO dynamics but has an enormous socio-economical relevance related with the numerous impacts associated with myLN events. After presenting some statistical evidence that myLN are not usually preceded by strong ENSO events, the authors propose a hypothesis for an alternative myLN precursor that involve another known interannual mode of variability of the region, the North Pacific Meridional Mode (PMM). While the author presents a simple but compelling statistical analysis that the role of strong ENSO as precursor of myLN events might have been overemphasized, they did not fully convince me that negative PMM is an actual myLN precursor. They need to provide evidence for the chain of events that might result in feedback between the westward extension of the negative SST anomalies associated with LN and the triggering of a negative PMM event. The manuscript, which is clear, well-written and with convincing arguments supported by simple but well-posed analysis, improve our understanding regarding myLN events and more broadly of ENSO dynamics. However, I have two main comments and few suggestions I would like to the authors to address before the manuscript will be suitable for publication with Nature Communications.

Response: We would like to express our appreciation to the reviewer for taking the time to evaluate our manuscript. The insightful comments and constructive suggestions offered by the reviewer have greatly enhanced the quality of our manuscript. In response to the reviewer's comments, we have performed additional analyses addressing the two main comments, namely comment 1 (the PMM and ENSO removal) and comment 2 (the chain of events on LN-PMM connection). We have also appropriately incorporated all other comments and suggestions raised by the reviewer into the revised manuscript. Our detailed point-by-point responses to the reviewer are as follows.

Major comments:

1. PMM and ENSO removal

The extract the variability associated with the PMM, an index tracking ENSO variability is generally regressed out. In the original work by Chiang and Vimont (2004)

the PMM is obtained by linearly removing (regressing out) the Cold Tongue Index, and nowadays a common choice is to remove the Niño3.4 index. This step is crucial as the dominance of ENSO variability in tropical Pacific SSTs would covers this partially ENSO independent variability in the North subtropical Pacific that we refer to as PMM. I would like to see the SST/Wind maps shown in the main text also in this “ENSO removed space” as it is crucial to distinguish between ENSO decay dynamics and actual PMM variability.

Response: We thank the reviewer for the insightful comment. Following the reviewer’s suggestion, we conducted further analysis on the SST/Wind maps presented in the main text (refer to Fig. 2a-c in the original manuscript), with a specific focus on the “**ENSO removed space**”. As suggested by the reviewer, the ENSO removed space (i.e., ENSO removal) was achieved by regressing out the Niño3.4 index from SST/Wind fields prior to analysis. The results shown in Figure B1 here indicate that the defining characteristics of a negative PMM (i.e., cold SST and accompanying northeasterly anomalies in the subtropical North Pacific) are detected in both multi-year La Niña groups: myLN_wPrecEN and myLN_w/oPrecEN (Fig. B1a and B1b). In contrast, these characteristics are not observed in the single-year La Niña group: syLN (Fig. B1c). These are overall consistent with the original results obtained with no ENSO removal.

Nevertheless, it is also found that the intensities of the negative PMM are somewhat weakened in the ENSO removed space when compared to the original results (Fig. B1 vs Fig. 2a-c in the original manuscript). Table B1, which displays the averaged PMM index values for myLN_wPrecEN and myLN_w/oPrecEN, quantitates this weakening by showing that the negative PMM intensities with ENSO removal are approximately 25% lower compared to those with no ENSO removal (–26% for myLN_wPrecEN and –23% for myLN_w/oPrecEN; compare the values in the 3rd row with those in the 2nd row in Table B1). Based on the findings above, we can speculate while the PMM is partially independent of ENSO, it is also dependent on ENSO to some extent, thus being influenced by it. The detailed influence of ENSO on the PMM has been reported in previous studies (Stuecker, 2018; Fang & Yu, 2020; Ding et al., 2022; Kim & Yu, 2022) and is also proposed in this study, which suggests that the negative PMM is triggered when the cold anomalies of the 1st-year La Niña extend westward in the tropical central Pacific (refer to the Sect. “Role of negative PMM: Crucial” in the original manuscript).

The influence of ENSO on the PMM, as proposed in this study, receives further support from the results presented in Table B1. Specifically, the weakening of negative PMM intensities is greater when conducting ENSO removal using the **Niño4 index** (SST anomaly averaged in the **tropical central Pacific**) compared to the use of the **Niño3.4 index** (SST anomaly averaged in the **tropical central-eastern Pacific**). Conversely, when using the **Niño3 or Niño1+2 indices** (SST anomalies averaged in the **tropical eastern or far-eastern Pacific**, respectively), the reduction rates of negative PMM intensities are lower than those using the Niño3.4 index. These results indicate that when a La Niña event extends further westward into the tropical central Pacific, rather than into the tropical eastern or far-eastern Pacific, its influence on the occurrence of negative PMM becomes stronger.

In response, we have incorporated discussions of these points in the revised manuscript (Lines 244-258 and 259-268). Figure B1 and Table B1, here, have been added as Supplementary Figure 7 and Table 2 in the revised manuscript, respectively.

Refs.

Stuecker, M. F. (2018). Revisiting the Pacific meridional mode. *Scientific reports*, 8(1), 3216.

Fang, S. W., & Yu, J. Y. (2020). A control of ENSO transition complexity by tropical Pacific mean SSTs through tropical-subtropical interaction. *Geophysical Research Letters*, 47(12), e2020GL087933.

Ding, R., Tseng, Y. H., Di Lorenzo, E., Shi, L., Li, J., Yu, J. Y., ... & Li, F. (2022). Multi-year El Niño events tied to the North Pacific Oscillation. *Nature communications*, 13(1), 3871.

Kim, J. W., & Yu, J. Y. (2022). Single-and multi-year ENSO events controlled by pantropical climate interactions. *npj Climate and Atmospheric Science*, 5(1), 88.

Fig. B1 | Same as Fig. 2a-c in the original manuscript except with ENSO signals removed by regressing out the Niño3.4 index from SST and wind fields prior to analysis

Table B1 | PMM index values (in s.d.) during the second spring (March¹-May¹) are shown for both multi-year La Niña groups: myLN_wPrecEN and myLN_w/oPrecEN. The values are presented with no ENSO removal (2nd row) and with ENSO removal using Niño3.4, Niño4, Niño3, and Niño1+2 indices (3rd to 6th row, respectively). Reduction rates, expressed as percentages relative to the values in the 2nd row, are denoted in parentheses.

PMM index during March ¹ -May ¹	myLN_wPrecEN	myLN_w/oPrecEN
No ENSO removal	-1.21	-1.29
ENSO removal using Niño3.4 index	-0.89 (-26%)	-0.99 (-23%)
ENSO removal using Niño4 index	-0.48 (-60%)	-0.72 (-44%)
ENSO removal using Niño3 index	-1.10 (-9%)	-1.09 (-16%)
ENSO removal using Niño1+2 index	-1.20 (-1%)	-1.23 (-5%)

2. LN-PMM-LN Dynamics: Chain of events

Perhaps the main contribution of this study is the proposed chain of events that results in feedback between LN and PMM. The authors put forward an interesting hypothesis for this LN-PMM connection, that is explained in the text with the following lines:

These westward-extended cold anomalies can cause negative diabatic heating anomalies in the tropical central Pacific (165°E to 165°W) by readily reducing SSTs below the convective threshold temperature, which suppresses deep convection in that region. This suppression excites stationary atmospheric Rossby waves into the North Pacific, inducing a subtropical anticyclone anomaly near Hawaii within the PMM region.

However, despite the fundamental role played by these dynamics in supporting author's claims, this chain of events is not proved but left as speculations. I think the authors needs to prove what they wrote in the above lines by performing an additional set of analysis focusing on diabatic heating, convective activity, Rossby waves propagation,

and sea-level anomalies. Also, looking at the cross-correlation function might help pinning down causality relationships.

Response: We thank the reviewer for the insightful comment. In response to the reviewer's suggestion, we have conducted additional analyses to provide further evidence on the detailed dynamics for the LN-PMM connection. These analyses involved the inclusion of two additional data: (1) Precipitation (Pr) data, which focuses on diabatic heating in the tropical central Pacific, and (2) mean sea level pressure (SLP) data, which focuses on atmospheric teleconnection over the North Pacific. To cover our analysis period from 1900 to 2022, we utilized two datasets: the NOAA 20CRv2 data (Compo et al., 2011) for the period 1900–1947, and NCEP/NCAR R1 data (Kalnay et al., 1996) for the period 1948–2022.

Regarding the first process in the connection between the westward-extended cold anomalies and negative diabatic heating anomalies in the tropical central Pacific (as mentioned in the comment: *“These westward-extended cold anomalies can cause negative diabatic heating anomalies in the tropical central Pacific (165°E to 165°W) by readily reducing SSTs below the convective threshold temperature, which suppresses deep convection in that region.”*), Figure B2, which illustrates the relationship between the TCP-SST index and TCP-Pr index (refer to the figure caption for index definitions), provides support for this connection by demonstrating the following:

- (i) There exists a strong positive correlation ($r=0.63$, $p=0.007$) between SST and precipitation (diabatic heating) anomalies in the tropical central Pacific during the first winter of La Niña events.
- (ii) Multi-year La Niña events, regardless of their group, are characterized by consistently lower SST values, which fall below the convective threshold of 28°C (indicated by the green line). These events exhibit stronger negative precipitation anomalies in the tropical central Pacific compared to single-year events (depicted by black dots vs. gray dots). The averaged values and spreads of the multi-year and single-year La Niña groups demonstrate that the differences in SST (as TCP-SST index) and precipitation anomalies (as TCP-Pr index) are statistically significant between the two groups.

Regarding the second process in the connection between the suppression in the tropical central Pacific and the formation of the subtropical anticyclone anomaly (as mentioned in the

comment: “*This suppression excites stationary atmospheric Rossby waves into the North Pacific, inducing a subtropical anticyclone anomaly near Hawaii within the PMM region.*”), we would like to begin by presenting a composite analysis of anomalous SLP and surface wind for three La Niña groups during the transition season between the first winter and the second spring, from January¹ to April¹ (Fig. B3). The composite maps provide supporting evidence for our claim as they illustrate that the subtropical anticyclone anomaly is exclusively observed in the multi-year La Niña groups (Fig. B3a, b) but not in the single-year La Niña group (Fig. B3c). This subtropical anticyclone anomaly, with its southern flank positioned to the east of Hawaii within the PMM region (indicated by the parallelograms in Fig. B3), plays a crucial role in the occurrence of a negative PMM during the subsequent spring. Next, as suggested by the reviewer, we have performed a correlation analysis to examine the causality between the suppression in the tropical central Pacific and the formation of the subtropical anticyclone anomaly. Figure B4 presents a lagged correlation map of SLP and surface wind anomalies during January¹–April¹ against the TCP-Pr index (multiplied by -1) during the preceding winter of December⁰–February¹. The result clearly demonstrates that suppression in the tropical central Pacific (as negative TCP-Pr index values) can induce a subtropical anticyclone anomaly (as positive SLP/clockwise wind correlations) in the subtropical North Pacific near the PMM region. Therefore, the correlation analysis further supports our claim in this study.

In response, we have revised our original writing in the revised manuscript to appropriately incorporate the points mentioned above (Lines 195-210 and 226-228). Figures B2-B4 have been newly added to the revised Supplementary Information as Supplementary Figures 3-5.

Refs.

Compo, G. P., Whitaker, J. S., Sardeshmukh, P. D., Matsui, N., Allan, R. J., Yin, X., ... & Worley, S. J. (2011). The twentieth century reanalysis project. *Quarterly Journal of the Royal Meteorological Society*, 137(654), 1-28.

Kalnay, E., Kanamitsu, M., Kistler, R., Collins, W., Deaven, D., Gandin, L., ... & Joseph, D. (1996). The NCEP/NCAR 40-year reanalysis project. *Bulletin of the American meteorological Society*, 77(3), 437-472.

Fig. B2 | Scatter plot of the TCP-SST index values against the TCP-Pr index values during the first winter for the three La Niña groups. The plot is similar to Fig. S3 in the original Supplementary Information except for the TCP-Pr index. The TCP-SST index was defined as raw SST averaged in the tropical central Pacific (5°S–5°N, 165°E–165°W). Similarly, the TCP-Pr index was defined as precipitation anomaly averaged in the tropical central Pacific, with its amplitude normalized by the standard deviation.

Fig. B3 | Composite structures of mean sea level pressure (SLP) anomalies (shading, in hPa) and surface wind (vector, in m s^{-1}) for the three La Niña groups during the transitioning season between first winter and spring (January¹ to April¹). The plot is similar to Fig. 2a-c in the original manuscript except for the SLP data.

Fig. B4 | Lagged correlation map of anomalous SLP (shading) and surface wind (vector) during January¹ to April¹ against the TCP-Pr index during December⁰ to February¹ for the period 1900–2022. Only areas displaying significant correlation coefficients at the 95% confidence level based on the Student’s t-test are shown. The TCP-Pr index is multiplied by -1 , so that its positive values correspond to negative diabatic heating anomalies. The black parallelogram delineates the region where a negative PMM occurs and exerts its influence.

Minor comment:

3. The authors refer to PMM but they are actually analyzing the NPMM. I agree that there is confusion in the literature as the original definition of PMM has somewhat implicitly evolved in NPMM after that Zhang et al., 2014 defined the South PMM (SPMM). I suggest replacing PMM in NPMM or at least informing the reader that PMM is known in literature also as NPMM.

Response: Thank you for the comment. After careful consideration of replacing PMM with NPMM, we have decided to retain PMM for the sake of brevity and to reduce potential confusion arising from the term “negative NPMM”, which is a shortened term referring to the negative phase of the NPMM. **However, as suggested by the reviewer, we have included a mention in the revised manuscript that the PMM used in this study is also referred to as the NPMM (Lines 23 and 73-74).**

4. Line 247: Given the reduced number of myLN event in the records, the authors use

data from 1900 to 2022, which contains highly unreliable decades (~1900-1950). The validity of the results for the first half of the 20th century should be discussed in the main text with a couple of sentences.

Response: Thank you for the comment. We acknowledge that there are concerns about the reliability of the selected La Niña events prior to the 1950s, presumably due to the unreliable climate observations and data assimilation techniques during this period. Therefore, careful interpretation is required for the La Niña events before the 1950s, and these events will be strictly validated in future studies. One approach could be to utilize proxy data, such as modern coral records or sediment cores, for comparison (e.g., An et al., 2012). **We have included a discussion of these points in the revised manuscript (Lines 395-400). The study of An et al. (2012) has been newly cited.**

Ref. An, S. I., Kim, J. W., Im, S. H., Kim, B. M., & Park, J. H. (2012). Recent and future sea surface temperature trends in tropical Pacific warm pool and cold tongue regions. *Climate dynamics*, 39, 1373-1383.

5. Line 262 and 277: expressed the threshold -0.5°C also in units of STD

Response: Thank you for the comment. **The value of -0.54 STD, which is equivalent to the threshold of -0.5°C , has been stated in Line 373 of the revised manuscript.**

6. Line 330: Codes should be available without request to the corresponding author. They can use zenodo.

Response: Thank you for the comment. **In response, the codes used in this study are now available at Zenodo (<https://zenodo.org/record/8221820>). We have made modification to Lines 461-462 in the revised manuscript.**

7. I think might be worth discussing the implications of authors' findings in the context of climate change and projected changes in PMM variability. Specifically, under anthropogenic forcing one might expect an increase in the variance of myLN events because of the robust projected increase in the variance of the PMM (Liguori and Di Lorenzo, 2018), which the authors' claim to be a precursor of myLN.

Liguori G. and Di Lorenzo E., 2018: Meridional Modes and Increasing Pacific decadal variability under greenhouse forcing. *Geophysical Research Letters*, 45. <https://doi.org/10.1002/2017GL076548>.

Response: Thank you for the comment. Based on the results of Liguori & Lorenzo (2018), we anticipate an increase in the frequency of multi-year La Niña events in the future. This increase is attributed to the robustly projected strengthening of the PMM variability as a result of anthropogenic forcing, which facilitates the activation of the PMM mechanism more easily. A recent paper by Geng et al. (2023), using all available CMIP6 model simulation scenarios, also supports this by revealing that the likelihood of multi-year La Niña events is significantly higher in the 21st century relative to the 20th century. These imply that the significance of the PMM's role in generating multi-year La Niña events could be further emphasized in future climate change. **We have addressed this implication in the revised manuscript (Lines 344-352). The studies of Liguori & Di Lorenzo (2018) and Geng et al. (2023) have been newly cited.**

Refs.

Liguori, G., & Di Lorenzo, E. (2018). Meridional modes and increasing Pacific decadal variability under anthropogenic forcing. *Geophysical Research Letters*, 45(2), 983-991.

Geng, T., Jia, F., Cai, W., Wu, L., Gan, B., Jing, Z., ... & McPhaden, M. J. (2023). Increased occurrences of consecutive La Niña events under global warming. *Nature*, 619(7971), 774-781.

REVIEWER COMMENTS

Reviewer #1 (Remarks to the Author):

Although the quality of the manuscript has been increased by addressing the reviewers' comments, my primary concern centers around the insufficient discussion of crucial findings from previous studies within the manuscript. Despite incorporating many directly relevant references in the current work, none of them have been thoroughly elucidated. This omission consequently undermines the novelty of this study. I am concerned about how this work could extend upon previous research and potentially advance our knowledge on the relationship between the Pacific Meridional Mode (PMM) and multi-year La Niña events (LNs). It's worth noting that this relationship seems to have been well-established in recent literature.

For instance, Studies 1,2 have explicitly explored the connections between the PMM and multi-year LNs. Notably, in the 'Discussion and Summary' section of Park et al. (2020) 1, they compared two mechanisms that could influence consecutive LN occurrences. The first mechanism involves discharge and recharge processes, while the second relates to wind stress variations associated with the PMM. They concluded that the PMM might lead to an inefficient recharging process, thereby contributing to the prolonged persistence of LNs. Interestingly, their conclusions appear to significantly overlap with the principal findings of the current study. However, a clear clarification on this matter is currently absent in the text. Furthermore, the recent work by Geng et al. (2023) 2 has highlighted the substantial role played by the (N)PMM in generating multi-year LNs by investigating CMIP6 models. Other studies 3–5 also suggested that the multi-year ENSO events are associated with PMM variability. I have noticed that, while most of these papers are duly cited in the text, regrettably, their primary findings and contributions have been overlooked. Considering this, I believe that a more comprehensive integration of these previously established insights would not only enrich the current manuscript's novelty but also enhance its significance within the broader scientific discourse.

Regarding my previous comment (General Comment#5 in the first-round review) on the role of positive PMM and multi-year ENs, the authors have responded that it exceeds the scope of the present work. However, I have recently come across one of the authors' previous papers 6, which clearly argued that the positive PMM is often induced by central Pacific El Niño events (ENs) and tends to lead to multi-year ENs via the wind-evaporation-SST (WES) feedback. Firstly, I do not see a reason for not referencing this previous work when addressing my concern. Secondly, both sets of findings seem to suggest that positive/negative PMMs can give rise to multi-year ENs/LNs through the WES feedback. Despite the phase difference, these findings seem remarkably parallel. Unless the text has addressed fundamentally distinct aspects that determine multi-year LNs, I am concerned

that the current results might lack comprehensiveness and novelty, especially if these findings could be extrapolated consistently across different phases of ENSO.

Regarding my previous comments on the South Pacific Meridional Mode (SPMM), the authors have asserted that the mechanisms linking the SPMM to ENSO are not as well-defined as those of the (N)PMM. This was precisely my intent in suggesting the inclusion of the SPMM. As the authors have noticed, the NPMM-ENSO mechanism is already firmly established, which significantly diminishes the novelty of this work. However, by incorporating the SPMM into the analysis, it would enhance the value of the research by providing a more comprehensive understanding of the meridional modes in both hemispheres and their roles in contributing to multi-year ENSO events.

The inclusion of the CESM1 analysis and the ENSO removal analysis undeniably bolsters the robustness of the key findings presented in this study. Nevertheless, my concerns remain predominantly centered around the aspect of novelty, largely due to the absence of a comprehensive engagement with prior literature. It's worth noting that the same CESM1 simulation has been exclusively employed and discussed in the authors' recent paper 6, addressing similar topics pertaining to multi-year ENSO events. Regrettably, there seems to be a lack of explicit comparison elucidating the distinctions between this newly added analysis and the prior investigations. Moreover, the utilization of ENSO removal analysis is widespread in meridional modes research. Notable examples include Chiang and Vimont (2004) 7, Chang et al. (2007) 8, Fig. 1a in Amaya (2019) 9, Zhang et al. (2014) 10, and You and Furtado (2017) 11. However, none of them have been referenced in the newly added 'ENSO removal' segment in the 'Methods' section.

References:

1. Park, J.-H. et al. Mid-latitude leading double-dip La Niña. *Int. J. Climatol.* 41, E1353–E1370 (2021).
2. Geng, T. et al. Increased occurrences of consecutive La Niña events under global warming. *Nature* 619, 774–781 (2023).
3. Ding, R. et al. Multi-year El Niño events tied to the North Pacific Oscillation. *Nat. Commun.* 13, 3871 (2022).
4. Fang, S.-W. & Yu, J.-Y. A Control of ENSO Transition Complexity by Tropical Pacific Mean SSTs Through Tropical-Subtropical Interaction. *Geophys. Res. Lett.* 47, e2020GL087933 (2020).
5. Fan, H., Wang, C. & Yang, S. Asymmetry Between Positive and Negative Phases of the Pacific Meridional Mode: A Contributor to ENSO Transition Complexity. *Geophys. Res. Lett.* 50, e2023GL104000 (2023).

6. Kim, J.-W. & Yu, J.-Y. Single- and multi-year ENSO events controlled by pantropical climate interactions. *Npj Clim. Atmospheric Sci.* 5, 1–11 (2022).
7. Chiang, J. C. H. & Vimont, D. J. Analogous Pacific and Atlantic Meridional Modes of Tropical Atmosphere–Ocean Variability*. *J. Clim.* 17, 4143–4158 (2004).
8. Chang, P. et al. Pacific meridional mode and El Niño–Southern Oscillation: PACIFIC MERIDIONAL MODE AND ENSO. *Geophys. Res. Lett.* 34, (2007).
9. Amaya, D. J. The Pacific Meridional Mode and ENSO: a Review. *Curr. Clim. Change Rep.* 5, 296–307 (2019).
10. Zhang, H., Clement, A. & Di Nezio, P. The South Pacific Meridional Mode: A Mechanism for ENSO-like Variability. *J. Clim.* 27, 769–783 (2014).
11. You, Y. & Furtado, J. C. The role of South Pacific atmospheric variability in the development of different types of ENSO: SOUTH PACIFIC OSCILLATION AND ENSO. *Geophys. Res. Lett.* 44, 7438–7446 (2017).

Reviewer #2 (Remarks to the Author):

I am satisfied with the modification made by the authors. I think the manuscript has improved substantially, thanks aslo to the excellent points made by the first reviewer.

Point-by-Point Responses to Reviewer #1

Although the quality of the manuscript has been increased by addressing the reviewers' comments, my primary concern centers around the insufficient discussion of crucial findings from previous studies within the manuscript. Despite incorporating many directly relevant references in the current work, none of them have been thoroughly elucidated. This omission consequently undermines the novelty of this study. I am concerned about how this work could extend upon previous research and potentially advance our knowledge on the relationship between the Pacific Meridional Mode (PMM) and multi-year La Niña events (LNs). It's worth noting that this relationship seems to have been well-established in recent literature.

Response: We sincerely appreciate the careful comments and suggestions provided by the reviewer during the second-round review. In this second revised manuscript, we have made further modifications to include a comprehensive discussion of the key findings from earlier studies regarding the relationship between the PMM and multi-year La Niña events. Additionally, we have added statements to highlight the novelty of our study by contrasting it with previous findings (see our replies to Comment 1). All other comments and suggestions raised by the reviewer have also been appropriately incorporated into the revised manuscript (Comments 2 to 4). Our detailed point-by-point responses to the reviewer's comments are as follows.

1. For instance, studies^{1,2} have explicitly explored the connections between the PMM and multi-year LNs. Notably, in the 'Discussion and Summary' section of Park et al. (2021)¹, they compared two mechanisms that could influence consecutive LN occurrences. The first mechanism involves discharge and recharge processes, while the second relates to wind stress variations associated with the PMM. They concluded that the PMM might lead to an inefficient recharging process, thereby contributing to the prolonged persistence of LNs. Interestingly, their conclusions appear to significantly overlap with the principal findings of the current study. However, a clear clarification on this matter is currently absent in the text. Furthermore, the recent work by Geng et al. (2023)² has highlighted the substantial role played by the (N)PMM in generating multi-year LNs by investigating CMIP6

models. Other studies³⁻⁵ also suggested that the multi-year ENSO events are associated with PMM variability. I have noticed that, while most of these papers are duly cited in the text, regrettably, their primary findings and contributions have been overlooked. Considering this, I believe that a more comprehensive integration of these previously established insights would not only enrich the current manuscript's novelty but also enhance its significance within the broader scientific discourse.

Response: Thank you for the comment. We would firstly like to provide a summary of the findings from previous studies, including those by Park et al. (2021) and Geng et al. (2023) mentioned in the comment. Their findings on the generation mechanism of multi-year La Niña events can be encapsulated as follows:

- (1) Regarding tropical ENSO dynamics: Multi-year La Niña events tend to follow a strong El Niño, commonly characterized by its maximum SST anomalies towards the equatorial eastern Pacific (Niño3 index > Niño4 index). The strong El Niño induces a large discharge of upper-ocean heat content in the tropical Pacific, which requires more than one La Niña to recharge to the climatological level.
- (2) Regarding subtropical ENSO dynamics: Multi-year La Niña events also exhibit a robust connection to mid-latitudes through the PMM in the subtropical North Pacific. During the developing phase of the first La Niña in boreal spring, a negative PMM often emerges as a result of the Gill-type atmospheric response to the strong El Niño SST anomalies over the equatorial eastern Pacific (Gill, 1980; Fang & Yu, 2020). This negative PMM persists in subsequent seasons due to thermodynamic air-sea couplings over the subtropical North Pacific, favoring the development of a La Niña event in winter, characterized by a meridionally wider pattern of cold SST and easterly wind anomalies. This meridionally broad La Niña is accompanied by a weaker negative wind stress curl at more extratropical latitudes, which leads to a slower heat recharge during La Niña due to reduced equatorward Sverdrup transport.
- (3) The aforementioned tropical and subtropical dynamical processes (1) and (2) both contribute to the persistence of cold SST anomalies throughout the decaying phase of the first La Niña into spring. These anomalies can be further strengthened by the seasonal positive Bjerknes feedback in late summer and autumn, which triggers a second La Niña, ultimately leading to the formation of multi-year La Niña events.

Next, we want to highlight the uniqueness of our current study, stemming from our challenge to conventional perspectives. To recap, prior research has predominantly focused on the influence of El Niño's intensity in the previous year, particularly during a strong El Niño, in shaping multi-year La Niña events. Their examination of the PMM in relation to multi-year La Niña revolves around its role in decelerating the recharge process triggered by the initial-year La Niña. As a result, the PMM's role has been considered merely a secondary adjustment to the primary discharge-recharge process associated with the preceding strong El Niño. In contrast, our study places direct emphasis on the subtropical ENSO dynamics linked to the PMM as the foremost and crucial mechanism in generating multi-year La Niña events. Our research presents perspectives and findings that are notably distinct and divergent from those found in earlier studies.

In more detail, the analyses in this study clearly demonstrate that the majority of multi-year La Niña events, both in observations (64%) and in a climate model simulation (61%), do not necessarily require a preceding strong El Niño. These findings indicate that the mechanisms proposed in prior studies do not fully account for a substantial portion of multi-year La Niña events (referred to as myLN_w/oPrSEN events in this study), as they inherently assume that a preceding strong El Niño is a key ingredient. However, the PMM mechanism we identified in this study, involving two-way interactions between ENSO (first La Niña) and PMM (negative PMM following the first La Niña), can operate independently without the need for the presence of a preceding strong El Niño. Consequently, it provides a comprehensive explanation for the generation of multi-year La Niña events, regardless of the El Niño conditions in their preceding years. These findings not only contribute to the advancement of our understanding of ENSO dynamics but also carry substantial socio-economic relevance, given the prolonged climate impacts associated with multi-year La Niña events.

In the revised manuscript, we have included more extensive discussions of recent primary findings on the relationship between the PMM and multi-year La Niña events, with additional emphasis on the novelty of our own findings (Lines 68-90 and 319-321). The previous studies mentioned in the comment, including other relevant ones, have also been appropriately referenced.

Refs.

Park, J. H., An, S. I., Kug, J. S., Yang, Y. M., Li, T., & Jo, H. S. (2021). Mid-latitude leading double-dip La Niña. *International Journal of Climatology*, 41, E1353-E1370.

Geng, T., Jia, F., Cai, W., Wu, L., Gan, B., Jing, Z., ... & McPhaden, M. J. (2023). Increased occurrences of consecutive La Niña events under global warming. *Nature*, 619(7971), 774-781.

Gill, A. E. (1980). Some simple solutions for heat-induced tropical circulation. *Quarterly Journal of the Royal Meteorological Society*, 106(449), 447-462.

Fang, S. W., & Yu, J. Y. (2020). A Control of ENSO Transition Complexity by Tropical Pacific Mean SSTs through Tropical-Subtropical Interaction. *Geophysical Research Letters*, e2020GL087933.

2. Regarding my previous comment (General Comment#5 in the first-round review) on the role of positive PMM and multi-year ENs, the authors have responded that it exceeds the scope of the present work. However, I have recently come across one of the authors' previous papers 6, which clearly argued that the positive PMM is often induced by central Pacific El Niño events (ENs) and tends to lead to multi-year ENs via the wind-evaporation-SST (WES) feedback. Firstly, I do not see a reason for not referencing this previous work when addressing my concern. Secondly, both sets of findings seem to suggest that positive/negative PMMs can give rise to multi-year ENs/LNs through the WES feedback. Despite the phase difference, these findings seem remarkably parallel. Unless the text has addressed fundamentally distinct aspects that determine multi-year LNs, I am concerned that the current results might lack comprehensiveness and novelty, especially if these findings could be extrapolated consistently across different phases of ENSO.

Response: Thank you for the comment. As the reviewer pointed out, we recognize the studies exploring the relationships between a positive PMM and multi-year El Niño events, such as the works of Ding et al. (2022) and Kim & Yu (2022). These studies specifically investigated how a positive PMM contributes to the occurrence of multi-year El Niño events using both observations and climate model simulations. Similar to the mechanism associated with a negative PMM, these studies have reported a mechanism involving a positive PMM, which is often induced by central Pacific El Niño events and is prone to leading to multi-year El Niño events. The summary of their findings is as follows:

(1) Ding et al. (2022) identified the positive phase of the North Pacific Oscillation (NPO) as a

crucial factor in the generation of multi-year El Niño events. The NPO signifies an intrinsic extratropical atmospheric variability, manifesting as a north-south seesaw in wintertime sea level pressure anomalies over the North Pacific. During winter, the positive NPO forcing effectively weakens the northeasterly trade winds and warms the SSTs in the subtropical northeastern Pacific. This, in turn, triggers a positive PMM during the succeeding spring. This positive PMM activates the wind-evaporation-SST feedback mechanism, leading to the extension of warm SST anomalies equatorward into the central equatorial Pacific. As a result, a Central Pacific (CP)-type El Niño event develops during the subsequent winter. This CP-type El Niño can excite atmospheric teleconnections to the extratropics, re-energizing the NPO variability and inducing another episode of positive NPO. This cyclic process re-activates the positive PMM and re-triggers the development of another El Niño, resulting in a multi-year El Niño event.

- (2) Kim & Yu (2022) also underscored the role of positive PMM as a crucial factor for the generation of a multi-year El Niño event. While the underlying physical mechanisms are broadly akin to those delineated in Ding et al. (2022), their emphasis laid on the importance of PMM-involved intra-basin interactions within the same Pacific basin between tropics and subtropics, rather than solely focusing on the role of NPO variability.

Regarding the novelty of our study, we kindly request the reviewer to refer to our prior response to Comment 1. Furthermore, it is important to highlight that the subtropical ENSO dynamics associated with the PMM exhibit a more pronounced El Niño-La Niña asymmetry compared to the conventional tropical ENSO dynamics linked to the recharge-discharge process (e.g., Yu & Fang, 2018; Fan et al., 2023). Consequently, the conclusions presented by Ding et al. (2022) and Kim & Yu (2022) concerning the positive PMM and multi-year El Niño may not necessarily extrapolate to the negative PMM and multi-year La Niña. Studies like our current research, which explore whether these mechanisms operate differently in the context of positive/negative PMM and multi-year El Niño/La Niña, hold significant value and merit recognition for their novelty.

In response, we have added a statement in the revised manuscript that a similar mechanism has been reported by Ding et al. (2022) and Kim & Yu (2022) (Lines 237-239). Further details, including the summary of their findings and our response regarding the novelty of this study, can be found in Supplementary Text 1.

Refs.

Ding, R., Tseng, Y. H., Di Lorenzo, E., Shi, L., Li, J., Yu, J. Y., ... & Li, F. (2022). Multi-year El Niño events tied to the North Pacific Oscillation. *Nature communications*, 13(1), 3871.

Kim, J. W., & Yu, J. Y. (2022). Single-and multi-year ENSO events controlled by pantropical climate interactions. *npj Climate and Atmospheric Science*, 5(1), 88.

Yu, J. Y., & Fang, S. W. (2018). The distinct contributions of the seasonal footprinting and charged-discharged mechanisms to ENSO complexity. *Geophysical Research Letters*, 45(13), 6611-6618.

Fan, H., Wang, C., & Yang, S. (2023). Asymmetry between positive and negative phases of the Pacific Meridional Mode: A contributor to ENSO transition complexity. *Geophysical Research Letters*, 50(14), e2023GL104000.

3. Regarding my previous comments on the South Pacific Meridional Mode (SPMM), the authors have asserted that the mechanisms linking the SPMM to ENSO are not as well-defined as those of the (N)PMM. This was precisely my intent in suggesting the inclusion of the SPMM. As the authors have noticed, the NPMM-ENSO mechanism is already firmly established, which significantly diminishes the novelty of this work. However, by incorporating the SPMM into the analysis, it would enhance the value of the research by providing a more comprehensive understanding of the meridional modes in both hemispheres and their roles in contributing to multi-year ENSO events.

Response: Thank you for the comment. As previously addressed in our response to Comment 6 during the first-round review, the inclusion of the SPMM in this study requires extensive additional analysis and is a completely different subject from the objects of our present study. Therefore, we continue to believe that it is unnecessary and goes beyond the intended scope of this study.

More importantly, we respectfully disagree with the reviewer's assertion that "**the NPMM-ENSO mechanism is already firmly established, which significantly diminishes the novelty of this work.**" As we have expounded upon in our responses to Comments 1 and 2 above, the primary and critical role of NPMM in driving multi-year La Niña events, along with its intricate mechanisms, has been largely overlooked in previous studies due to the

excessive emphasis on the recharge-discharge process. The mechanism involving NPMM and multi-year La Niña was also explored within this prevailing recharge-discharge framework. Our current study breaks away from this conventional perspective and meticulously illustrates how the NPMM-multi-year La Niña mechanism can operate independently of the recharge-discharge paradigm. We firmly believe that our findings, rooted in the investigation of the NPMM, can have a significant impact on subsequent research into multi-year La Niña dynamics. Furthermore, we maintain that the suggested examination of the SPMM-multi-year La Niña mechanism falls beyond the scope of our present study and is not an essential component of this manuscript.

4. The inclusion of the CESM1 analysis and the ENSO removal analysis undeniably bolsters the robustness of the key findings presented in this study. Nevertheless, my concerns remain predominantly centered around the aspect of novelty, largely due to the absence of a comprehensive engagement with prior literature. It's worth noting that the same CESM1 simulation has been exclusively employed and discussed in the authors' recent paper 6, addressing similar topics pertaining to multi-year ENSO events. Regrettably, there seems to be a lack of explicit comparison elucidating the distinctions between this newly added analysis and the prior investigations. Moreover, the utilization of ENSO removal analysis is widespread in meridional modes research. Notable examples include Chiang and Vimont (2004) 7, Chang et al. (2007) 8, Fig. 1a in Amaya (2019) 9, Zhang et al. (2014) 10, and You and Furtado (2017) 11. However, none of them have been referenced in the newly added 'ENSO removal' segment in the 'Methods' section.

Response: Thank you for the suggestion. In response, we have added a note in the revised manuscript that the CESM1 simulation has been exclusively employed and discussed in our previous study, Kim & Yu (2022), where the emphasis is on the controlling role of pantropical climate interactions (involving inter-basin interactions between the Pacific, Indian, and Atlantic Oceans and intra-basin interactions within the tropical and subtropical Pacific Oceans) in the formation of single- and multi-year ENSO events (Lines 328-333). Additionally, we have referenced the following studies in the 'ENSO removal' section of the Methods (Lines 450-451): Chiang & Vimont (2004), Chang et al. (2007), Zhang et al. (2014), You & Furtado (2017), and Amaya (2019).

Once again, we want to emphasize that our study identified the NPMM-multi-year La Niña mechanism to challenge the conventional multi-year La Niña mechanism based on the recharge-discharge framework. This research focus was not addressed in the CESM1 study conducted by Kim & Yu (2022). It is important to note that although both our current study and Kim & Yu (2022) utilized the same CESM1 simulation, their research objectives are inherently distinct, and both contribute unique findings to the field.

Refs.

Kim, J. W., & Yu, J. Y. (2022). Single-and multi-year ENSO events controlled by pantropical climate interactions. *npj Climate and Atmospheric Science*, 5(1), 88.

Chiang, J. C., & Vimont, D. J. (2004). Analogous Pacific and Atlantic meridional modes of tropical atmosphere–ocean variability. *Journal of Climate*, 17(21), 4143-4158.

Chang, P., Zhang, L., Saravanan, R., Vimont, D. J., Chiang, J. C., Ji, L., ... & Tippett, M. K. (2007). Pacific meridional mode and El Niño—Southern oscillation. *Geophysical Research Letters*, 34(16).

Zhang, H., Clement, A., & Di Nezio, P. (2014). The South Pacific meridional mode: A mechanism for ENSO-like variability. *Journal of Climate*, 27(2), 769-783.

You, Y., & Furtado, J. C. (2017). The role of South Pacific atmospheric variability in the development of different types of ENSO. *Geophysical Research Letters*, 44(14), 7438-7446.

Amaya, D. J. (2019). The Pacific meridional mode and ENSO: A review. *Current Climate Change Reports*, 5, 296-307.

References:

1. Park, J.-H. et al. Mid-latitude leading double-dip La Niña. *Int. J. Climatol.* 41, E1353–E1370 (2021).

2. Geng, T. et al. Increased occurrences of consecutive La Niña events under global warming. *Nature* 619, 774–781 (2023).

3. Ding, R. et al. Multi-year El Niño events tied to the North Pacific Oscillation. *Nat. Commun.* 13, 3871 (2022).

4. Fang, S.-W. & Yu, J.-Y. A Control of ENSO Transition Complexity by Tropical Pacific Mean SSTs Through Tropical-Subtropical Interaction. *Geophys. Res. Lett.* 47, e2020GL087933 (2020).

5. Fan, H., Wang, C. & Yang, S. Asymmetry Between Positive and Negative Phases of the Pacific Meridional Mode: A Contributor to ENSO Transition Complexity. *Geophys.*

Res. Lett. 50, e2023GL104000 (2023).

6. Kim, J.-W. & Yu, J.-Y. Single- and multi-year ENSO events controlled by pantropical climate interactions. *Npj Clim. Atmospheric Sci.* 5, 1–11 (2022).

7. Chiang, J. C. H. & Vimont, D. J. Analogous Pacific and Atlantic Meridional Modes of Tropical Atmosphere–Ocean Variability*. *J. Clim.* 17, 4143–4158 (2004).

8. Chang, P. et al. Pacific meridional mode and El Niño–Southern Oscillation: PACIFIC MERIDIONAL MODE AND ENSO. *Geophys. Res. Lett.* 34, (2007).

9. Amaya, D. J. The Pacific Meridional Mode and ENSO: a Review. *Curr. Clim. Change Rep.* 5, 296–307 (2019).

10. Zhang, H., Clement, A. & Di Nezio, P. The South Pacific Meridional Mode: A Mechanism for ENSO-like Variability. *J. Clim.* 27, 769–783 (2014).

11. You, Y. & Furtado, J. C. The role of South Pacific atmospheric variability in the development of different types of ENSO: SOUTH PACIFIC OSCILLATION AND ENSO. *Geophys. Res. Lett.* 44, 7438–7446 (2017).

Point-by-Point Responses to Reviewer #2

I am satisfied with the modification made by the authors. I think the manuscript has improved substantially, thanks also to the excellent points made by the first reviewer.

Response: We greatly appreciate the reviewer's comment!